# AdjointDPM: Adjoint Sensitivity Method for Gradient Backpropagation of Diffusion Probabilistic Models

**Jiachun Pan**[*]
National University of Singapore
pan.jc@nus.edu.sg

**Jun Hao Liew**
ByteDance
junhao.liew@bytedance.com

**Vincent Y. F. Tan**
National University of Singapore
vtan@nus.edu.sg

**Jiashi Feng**
ByteDance
jshfeng@bytedance.com

**Hanshu Yan**[*][†]
ByteDance
hanshu.yan@bytedance.com

## Abstract

This paper considers a ubiquitous problem underlying several applications of DPMs, *i.e.*, optimizing the parameters of DPMs when the objective is a differentiable metric defined on the generated contents. Since the sampling procedure of DPMs involves recursive calls to the denoising UNet, naïve gradient backpropagation requires storing the intermediate states of all iterations, resulting in extremely high memory consumption. To overcome this issue, we propose a novel method AdjointDPM, which first generates new samples from diffusion models by solving the corresponding probability-flow ODEs. It then uses the adjoint sensitivity method to backpropagate the gradients of the loss to the models' parameters (including conditioning signals, network weights, and initial noises) by solving another augmented ODE. To reduce numerical errors in both the forward generation and gradient backpropagation processes, we further reparameterize the probability-flow ODE and augmented ODE as simple non-stiff ODEs using exponential integration. AdjointDPM can effectively compute the gradients of all types of parameters in DPMs, including the network weights, conditioning text prompts, and noisy states. Finally, we demonstrate the effectiveness of AdjointDPM on several interesting tasks: guided generation via modifying sampling trajectories, finetuning DPM weights for stylization, and converting visual effects into text embeddings.[1]

## 1 Introduction

Diffusion Probabilistic Models (DPMs) constitute a family of generative models that diffuse data distributions into white Gaussian noise and then revert the stochastic diffusion process to synthesize new contents (Ho et al., 2020; Song et al., 2021). DPM-based methods have recently achieved state-of-the-art performances in generating various types of contents, such as images (Saharia et al., 2022; Rombach et al., 2022; Ramesh et al., 2022), videos (Blattmann et al., 2023; Zhou et al., 2022; Ho et al., 2022), and audio data (Liu et al., 2023a; Schneider, 2023). To promote the development of downstream tasks, several pre-trained high-performance models, such as Stable Diffusion (SD) (Rombach et al., 2022), have been made publicly available. Based on these public large-scale models, researchers have developed many algorithms for creative applications (Gal et al., 2022; Daras & Dimakis, 2022; Kawar et al., 2023; Ruiz et al., 2022; Fei et al., 2023; Wen et al., 2023;

---

[*]Equal contribution. This work was completed during Jiachun Pan's internship at ByteDance.
[†]Project Lead.
[1]Github link for codes: https://github.com/HanshuYAN/AdjointDPM.git

Molad et al., 2023). For example, a line of customization algorithms of DPMs, such as Textural-Inversion (Gal et al., 2022) and DreamBooth (Ruiz et al., 2022), have been proposed to adapt DPMs for generating images/videos that share certain styles or identities. Researchers also have proposed some guidance algorithms (Bansal et al., 2023; Ho & Salimans, 2021) to make the generation process more controllable.

A ubiquitous problem in customization and guidance applications is the optimization of the diffusion models' parameters so that the final generated contents satisfy certain properties. For example, to customize models for certain styles, we need to optimize the model weights to minimize the style distance between the generated images and the reference. Alternatively, concerning the guidance of sampling, we need to adjust the intermediate noisy states via the gradients of the guidance loss computed on the final generated data. Generally, the parameters to optimize include the conditional text embeddings, the network weights, and the noisy states, as they all affect the sampling trajectories. We formulate this optimization problem as follows. Denote the DPM as $\Phi(\cdot, \cdot, \epsilon_\theta)$, which generates samples by iteratively calling a function $\epsilon_\theta$. The desired properties can be defined by the loss function $L(\cdot)$ computed based on the generated contents $\mathbf{x}_0 = \Phi(\mathbf{x}_T, c, \epsilon_\theta)$. We aim to minimize the loss by optimizing the variables $\psi$, including the weights $\theta$, conditioning signals $c$, or initial noise $\mathbf{x}_T$, i.e.,

$$\min_{\psi \in \{\mathbf{x}_T, c, \theta\}} L(\Phi(\mathbf{x}_T, c, \boldsymbol{\epsilon}_\theta)). \tag{1}$$

To solve the optimization problem (1), an effective backpropagation (BP) technique is required to compute the gradient of the loss function $L(\mathbf{x}_0)$ with respect to the optimization variables. Song et al. (2021) showed that the DPM sampling process is equivalent to solving a probability-flow ODE. Thus, many efficient sampling methods have been developed using adaptive ODE solvers. The sampling process involves recursive calls to the denoising UNet $\epsilon_\theta(\mathbf{x}_t, t, c)$ for multiple iterations. Using naïve gradient BP requires intermediate state storage for all iterations, resulting in significant GPU memory consumption. To overcome this problem, we propose AdjointDPM, a novel gradient BP technique based on the adjoint sensitivity method (Chen et al., 2018). AdjointDPM computes the gradient by solving a backward ODE that only needs to store the intermediate state at the time point of function evaluation, resulting in constant memory usage. Moreover, we reparameterize the diffusion generation process to a simple non-stiff ODE using exponential integration, which helps reduce discretization errors in both the forward and reverse processes of gradient computation.

We evaluate the effectiveness of AdjointDPM by applying it to several interesting tasks, involving optimizing the initial/intermediate noisy states, network weights, and conditioning text prompts, respectively. 1) Guided sampling. Under the supervision of fine-grained vision classifiers, Adjoint-DPM can guide the Stable Diffusion to synthesize images of certain breeds of animals. 2) Security auditing of image generation systems. AdjointDPM successfully finds a set of initial noise whose corresponding output images contain NSFW (**n**ot **s**afe **f**or **w**ork) content, but can sneakily bypass the moderation filters. This triggers *an alert about the potential security issues of existing AI generation systems*. 3) Stylization via a single reference image. AdjointDPM can finetune a Stable Diffusion model for stylization defined by the Gram Matrix (Gatys et al., 2015) of a reference image. The stylizing capability of the fine-tuned model can generalize to different objects. All the empirical results demonstrate the flexibility and general applicability of AdjointDPM. We summarize the main contributions of this paper as follows:

1. We propose a novel gradient backpropagation method of DPMs by applying the adjoint sensitivity method to the sampling process of diffusion models.

2. To the best of our knowledge, AdjointDPM is the first general gradient backpropagation method that can be used for all types of parameters of DPMs, including network weights, conditioning text prompts, and intermediate noisy states.

3. AdjointDPM can be used for several creative applications and outperforms other baselines.

## 2 BACKGROUND

### 2.1 PROBABILITY-FLOW ODES CORRESPONDING TO DPMS

The framework of diffusion probabilistic models involves gradually diffusing the complex target data distribution to a simple noise distribution, such as white Gaussian, and solving the corresponding

reverse process to generate new samples (Ho et al., 2020; Song et al., 2021). Both the diffusion and denoising processes can be characterized by temporally continuous stochastic differential equations (SDEs) (Song et al., 2021). Song et al. (2021) derive deterministic processes (probability-flow ODEs) that are equivalent to the stochastic diffusion and denoising processes in the sense of marginal probability densities for all the time steps.

Let $q_0$ denote the unknown $d$-dimensional data distribution. Song et al. (2021) formulated the forward diffusion process $\{\mathbf{x}(t)\}_{t \in [0,T]}$ as follows

$$d\mathbf{x}_t = f(t)\mathbf{x}_t\, dt + g(t)\, d\mathbf{w}_t, \quad \mathbf{x}_0 \sim q_0(\mathbf{x}), \quad t \in [0,T], \tag{2}$$

where $\mathbf{x}_t$ denotes the state at time $t$ and $\mathbf{w}_t$ is the standard Wiener process. $f(t)\mathbf{x}_t$ is a vector-valued function called *drift* coefficient and $g(t)$ is a scalar function known as *diffusion* coefficient. For the forward diffusion process, it is common to adopt the conditional probability as $q_{0t}(\mathbf{x}_t|\mathbf{x}_0) = \mathcal{N}(\mathbf{x}_t|\alpha_t\mathbf{x}_0, \sigma_t^2\mathbf{I})$, and the marginal distribution of $\mathbf{x}_T$ to be approximately a standard Gaussian. For sampling, we have a corresponding reverse process as

$$d\mathbf{x}_t = [f(t)\mathbf{x}_t - g(t)^2\nabla_{\mathbf{x}_t}\log q_t(\mathbf{x}_t)]\, dt + g(t)\, d\mathbf{w}_t, \quad \mathbf{x}_T \sim \mathcal{N}(0, \sigma_T^2\mathbf{I}), \quad t \in [0,T]. \tag{3}$$

In Eqn. (3), the term $\nabla_{\mathbf{x}}\log q_t(\mathbf{x})$, is known as the *score function*. We can train a neural network $\epsilon_\theta(\mathbf{x}_t, t)$ to estimate $-\sigma_t\nabla_{\mathbf{x}}\log q_t(\mathbf{x}_t)$ via denoising score matching. As discussed in Song et al. (2021), there exists a corresponding deterministic process whose trajectory shares the same set of marginal probability densities $\{q_t(\mathbf{x})\}_{t=0}^T$ as the SDE (3). The form of this deterministic probability-flow ODE is shown in (4). We can generate new samples by solving Eqn. (4) from $T$ to 0 with initial sample $\mathbf{x}_T$ drawn from $\mathcal{N}(0, \sigma_T^2\mathbf{I})$.

$$d\mathbf{x} = \left[f(t)\mathbf{x}_t + \frac{g(t)^2}{2\sigma_t}\epsilon_\theta(\mathbf{x}_t, t)\right] dt \tag{4}$$

For conditional sampling, classifier-free guidance (CFG) (Ho & Salimans, 2021) has been widely used in various tasks for improving the sample quality, including text-to-image, image-to-image, class-to-image generation (Saharia et al., 2022; Dhariwal & Nichol, 2021; Nichol et al., 2022). We can use CFG to generate new samples by solving Eqn. (4) and replacing $\epsilon_\theta(\mathbf{x}_t, t)$ with $\tilde{\epsilon}_\theta(\mathbf{x}_t, t, c)$.

$$\tilde{\epsilon}_\theta(\mathbf{x}_t, t, c) := s \cdot \epsilon_\theta(\mathbf{x}_t, t, c) + (1-s) \cdot \epsilon_\theta(\mathbf{x}_t, t, \varnothing),$$

## 2.2 ADJOINT SENSITIVITY METHODS FOR NEURAL ODES

Considering a neural ODE model

$$\frac{d\mathbf{x}}{dt} = \mathbf{s}(\mathbf{x}_t, t, \theta),$$

the output $\mathbf{x}_0 = \mathbf{x}_T + \int_T^0 \mathbf{s}(\mathbf{x}_t, t, \theta)\, dt$. We aim to optimize the input $\mathbf{x}_T$ or the weights $\theta$ by minimizing a loss $L$ defined on the output $\mathbf{x}_0$. Regarding $\frac{\partial L}{\partial \mathbf{x}_T}$, Chen et al. (2018) introduced adjoint state $\mathbf{a}(t) = \frac{\partial L}{\partial \mathbf{x}_t}$, which represents how the loss w.r.t the state $\mathbf{x}_t$ at any time $t$. The dynamics of $\mathbf{a}(t)$ are given by another ODE,

$$\frac{d\mathbf{a}(t)}{dt} = -\mathbf{a}(t)^T\frac{\partial \mathbf{s}(\mathbf{x}_t, t, \theta)}{\partial \mathbf{x}_t}, \tag{5}$$

which can be thought of as the instantaneous analog of the chain rule. Since $\frac{\partial L}{\partial \mathbf{x}_0}$ is known, we can compute $\frac{\partial L}{\partial \mathbf{x}_T}$ by solving the initial value problem (IVP) backwards in time $T$ to 0 of ODE in (5). Similarly, for $\theta$, we can regard them as a part of the augmented state:

$$\frac{d}{dt}[\mathbf{x}, \theta, t](t) := [\mathbf{s}(\mathbf{x}_t, t, \theta), \mathbf{0}, 1].$$

The corresponding adjoint state to this augmented state are $\mathbf{a}_{\text{aug}}(t) := [\mathbf{a}(t), \mathbf{a}_\theta(t), \mathbf{a}_t(t)]$, where $\mathbf{a}_\theta := \frac{\partial L}{\partial \theta}$ and $\mathbf{a}_t := \frac{\partial L}{\partial t}$. The augmented adjoint state $\mathbf{a}_{\text{aug}}$ is governed by:

$$\frac{d\mathbf{a}_{\text{aug}}}{dt} = -\left[\mathbf{a}\frac{\partial \mathbf{s}}{\partial \mathbf{x}}, \mathbf{a}\frac{\partial \mathbf{s}}{\partial \theta}, \mathbf{a}\frac{\partial \mathbf{s}}{\partial t}\right]. \tag{6}$$

By solving the IVP from time $T$ to 0 of Eqn. (6), we obtain the gradients of $L$ w.r.t. $\{\mathbf{x}_t, \theta, t\}$. The explicit algorithm (Chen et al., 2018) is shown in Algorithm 1.

---

**Algorithm 1** Reverse-mode derivative of an ODE initial value problem

---

**Input:** Dynamics parameter $\theta$, start time $t_0$, end time $t_1$, final state $\mathbf{x}_{t_1}$, loss gradient $\partial L/\partial \mathbf{x}_{t_1}$.

$\quad a(t_1) = \frac{\partial L}{\partial \mathbf{x}_{t_1}}, a_\theta(t_1) = \mathbf{0}, z_0 = [\mathbf{x}_{t_1}, a(t_1), a_\theta(t_1)]$          ▷ Define initial augmented state.

$\quad$ **def** AugDynamics($[\mathbf{x}_t, \mathbf{a}_t, \cdot], t, \theta$)          ▷ Define dynamics on augmented state.

$\quad\quad$ **return** $[\mathbf{s}(\mathbf{x}_t, t, \theta), -\mathbf{a}_t^T \frac{\partial \mathbf{s}}{\partial \mathbf{x}}, -\mathbf{a}_t^T \frac{\partial \mathbf{s}}{\partial \theta}]$          ▷ Concatenate time-derivatives

$\quad [\mathbf{x}_{t_0}, \frac{\partial L}{\partial \mathbf{x}_{t_0}}, \frac{\partial L}{\partial \theta}] = \text{ODESolve}(z_0, \text{AugDynamics}, t_1, t_0, \theta)$          ▷ Solve reverse-time ODE

**Return:** $[\frac{\partial L}{\partial \mathbf{x}_{t_0}}, \frac{\partial L}{\partial \theta}]$          ▷ Return gradients

---

# 3 ADJOINT SENSITIVITY METHODS FOR DIFFUSION PROBABILISTIC MODELS

In this section, we develop the AdjointDPM for gradient backpropagation in diffusion models based on the adjoint sensitivity methods from the neural ODE domain. When optimizing the model's parameters $\mathbf{x}_T$ or $\theta$ (including the conditioning $c$), AdjointDPM first generates new samples via the forward probability-flow ODE (4). Through applying the adjoint sensitivity method, we then write out and solve the backward adjoint ODE (6) to compute the gradients of loss with respect to the parameters. One can apply any general-purpose numerical ODE solver, such as Euler–Maruyama and Runge–Kutta methods (Atkinson et al., 2011), for solving the ODE. To further improve the efficiency of the vanilla adjoint sensitivity methods, we exploit the semi-linear structure of the diffusion ODE functions (4), which has been used in several existing works for accelerating DPM samplers (Lu et al., 2022a;b; Karras et al., 2022; Zhang & Chen, 2022), and reparameterize the forward and backward ODEs as simple non-stiff ones.

## 3.1 APPLYING ADJOINT METHODS TO PROBABILITY-FLOW ODES

Sampling from DPMs, we obtain the generated data $\mathbf{x}_0 = \mathbf{x}_T + \int_T^0 \mathbf{s}(\mathbf{x}_t, t, \theta, c) \, \mathrm{d}t$, where

$$\mathbf{s}(\mathbf{x}_t, t, \theta, c) = f(t)\mathbf{x}_t + \frac{g(t)^2}{2\sigma_t} \tilde{\epsilon}_\theta(\mathbf{x}_t, t, c). \tag{7}$$

Concerning the customization or guidance tasks, we aim to minimize a loss $L$ defined on $\mathbf{x}_0$, such as the stylization loss or semantic scores. We plug the equation (7) into the augmented adjoint ODE (5), and obtain the reverse ODE function in Algorithm 1 as:

$$\mathrm{d}\begin{bmatrix} \mathbf{x}_t \\ \frac{\partial L}{\partial \mathbf{x}_t} \\ \frac{\partial L}{\partial \theta} \\ \frac{\partial L}{\partial t} \end{bmatrix} = -\begin{bmatrix} -f(t)\mathbf{x}_t - \frac{g(t)^2}{2\sigma_t}\tilde{\epsilon}_\theta(\mathbf{x}_t, t, c) \\ f(t)\frac{\partial L}{\partial \mathbf{x}_t} + \frac{\partial L}{\partial \mathbf{x}_t}\frac{g(t)^2}{2\sigma_t}\frac{\partial \tilde{\epsilon}_\theta(\mathbf{x}_t, t, c)}{\partial \mathbf{x}_t} \\ \frac{\partial L}{\partial \mathbf{x}_t}\frac{g(t)^2}{2\sigma_t}\frac{\partial \tilde{\epsilon}_\theta(\mathbf{x}_t, t, c)}{\partial \theta} \\ \frac{\mathrm{d}f(t)}{\mathrm{d}t}\frac{\partial L}{\partial \mathbf{x}_t}\mathbf{x}_t + \frac{\partial L}{\partial \mathbf{x}_t}\frac{\partial [g(t)^2/2\sigma_t\tilde{\epsilon}_\theta(\mathbf{x}_t, t, c)]}{\partial t} \end{bmatrix} \mathrm{d}t. \tag{8}$$

We observe that the ODEs governing $\mathbf{x}_t$ and $\frac{\partial L}{\partial \mathbf{x}_t}$ both contain linear and nonlinear parts. If we directly use off-the-shelf numerical solvers on Eqn. (8), it causes discretization errors of both the linear and nonlinear terms. To avoid this, in Section 3.2, we exploit the semi-linear structure of the probability-flow ODE to better control the discretization error for each step. Thus, we are allowed to use a smaller number of steps for generating samples of comparable quality.

## 3.2 EXPONENTIAL INTEGRATION AND REPARAMETERIZATION

We use the *exponential integration* to transform the ODE (4) into a simple non-stiff ODE. We multiply an integrating factor $\exp(-\int_0^t f(\tau)\mathrm{d}\tau)$ on both sides of Eqn. (4) and obtain

$$\frac{\mathrm{d}e^{-\int_0^t f(\tau)\mathrm{d}\tau}\mathbf{x}_t}{\mathrm{d}t} = e^{-\int_0^t f(\tau)\mathrm{d}\tau}\frac{g(t)^2}{2\sigma_t}\tilde{\epsilon}_\theta(\mathbf{x}_t, t, c).$$

Let $\mathbf{y}_t$ denote $e^{-\int_0^t f(\tau)\mathrm{d}\tau}\mathbf{x}_t$, then we have

$$\frac{\mathrm{d}\mathbf{y}_t}{\mathrm{d}t} = e^{-\int_0^t f(\tau)\mathrm{d}\tau}\frac{g(t)^2}{2\sigma_t}\tilde{\epsilon}_\theta\left(e^{\int_0^t f(\tau)\mathrm{d}\tau}\mathbf{y}_t, t, c\right). \tag{9}$$

We introduce a variable $\rho = \gamma(t)$ and $\frac{\mathrm{d}\gamma}{\mathrm{d}t} = e^{-\int_0^t f(\tau)\mathrm{d}\tau} \frac{g(t)^2}{2\sigma_t}$. In diffusion models, $\gamma(t)$ usually monotonically increases when $t$ increases from $0$ to $T$. For example, when we choose $f(t) = \frac{\mathrm{d}\log\alpha}{\mathrm{d}t}$ and $g^2(t) = \frac{\mathrm{d}\sigma_t^2}{\mathrm{d}t} - 2\frac{\mathrm{d}\log\alpha}{\mathrm{d}t}\sigma_t^2$ in VP-SDE (Song et al., 2021), we have $\gamma(t) = \alpha_0\frac{\sigma_t}{\alpha_t} - \sigma_0$. Thus, a bijective mapping exists between $\rho$ and $t$, and we can reparameterize (9) as:

$$\frac{\mathrm{d}\mathbf{y}}{\mathrm{d}\rho} = \tilde{\boldsymbol{\epsilon}}_\theta\left(e^{\int_0^{\gamma^{-1}(\rho)} f(\tau)\mathrm{d}\tau}\mathbf{y}, \gamma^{-1}(\rho), c\right). \tag{10}$$

We also reparameterize the reverse ODE function in Algorithm 1 as follows

$$\mathrm{d}\begin{bmatrix} \mathbf{y} \\ \frac{\partial L}{\partial \mathbf{y}} \\ \frac{\partial L}{\partial \theta} \\ \frac{\partial L}{\partial \rho} \end{bmatrix} = -\begin{bmatrix} -\tilde{\boldsymbol{\epsilon}}_\theta\left(e^{\int_0^{\gamma^{-1}(\rho)} f(\tau)\mathrm{d}\tau}\mathbf{y}, \gamma^{-1}(\rho), c\right) \\ \frac{\partial L}{\partial \mathbf{y}} \frac{\partial\tilde{\boldsymbol{\epsilon}}_\theta\left(e^{\int_0^{\gamma^{-1}(\rho)} f(\tau)\mathrm{d}\tau}\mathbf{y}, \gamma^{-1}(\rho), c\right)}{\partial \mathbf{y}} \\ \frac{\partial L}{\partial \mathbf{y}} \frac{\partial\tilde{\boldsymbol{\epsilon}}_\theta\left(e^{\int_0^{\gamma^{-1}(\rho)} f(\tau)\mathrm{d}\tau}\mathbf{y}, \gamma^{-1}(\rho), c\right)}{\partial \theta} \\ \frac{\partial L}{\partial \mathbf{y}} \frac{\partial\tilde{\boldsymbol{\epsilon}}_\theta\left(e^{\int_0^{\gamma^{-1}(\rho)} f(\tau)\mathrm{d}\tau}\mathbf{y}, \gamma^{-1}(\rho), c\right)}{\partial \rho} \end{bmatrix} \mathrm{d}\rho. \tag{11}$$

Now instead of solving Eqn. (4) and Eqn. (8), we use off-the-shelf numerical ODE solvers to solve Eqn. (10) and Eqn. (11). This method is termed AdjointDPM. Implementation details are provided in Appendix E.

## 3.3 Error Control

Here, we first show that the exact solutions of the reparameterzied ODEs are equivalent to the original ones. For the equation in the first row of Eqn. (11), its exact solution is:

$$\mathbf{y}_{\rho(t)} = \mathbf{y}_{\rho(s)} + \int_{\rho(s)}^{\rho(t)} \tilde{\boldsymbol{\epsilon}}_\theta\left(e^{\int_0^{\gamma^{-1}(\rho)} f(\tau)\mathrm{d}\tau}\mathbf{y}, \gamma^{-1}(\rho), c\right)\mathrm{d}\rho. \tag{12}$$

We can rewrite it as $e^{-\int_0^t f(\tau)\mathrm{d}\tau}\mathbf{x}_t = e^{-\int_0^s f(\tau)\mathrm{d}\tau}\mathbf{x}_s + \int_s^t \frac{\mathrm{d}\rho}{\mathrm{d}\tau}\tilde{\boldsymbol{\epsilon}}_\theta(\mathbf{x}_\tau, \tau, c)\mathrm{d}\tau$. Then, we have

$$\mathbf{x}_t = e^{\int_s^t f(\tau)\mathrm{d}\tau}\mathbf{x}_s + \int_s^t e^{\int_\tau^t f(r)\mathrm{d}r}\frac{g(\tau)^2}{2\sigma_\tau}\tilde{\boldsymbol{\epsilon}}_\theta(\mathbf{x}_\tau, \tau, c)\mathrm{d}\tau,$$

which is equivalent to the exact solution of the equation in the first row of Eqn. (8). Similarly, for other equations in (11), their exact solutions are also equivalent to the solutions in (8). Thus, when we numerically solve non-stiff ODEs in Eqns (10) and (11), there are only discretization errors for nonlinear functions and the closed form of integration of linear parts have been solved exactly without any numerical approximation.

In summary, we reformulate the forward and reverse ODE functions and show that by using off-the-shelf numerical ODE solvers on the reparameterized ODEs, AdjointDPM does not introduce discretization eeror to the linear part. In Section 3.4, we experimentally compare the FID of generated images by solving Eqn. (4) and Eqn. (10) with the same number of network function evaluations (NFE). The results verify the superiority of solving Eqn. (10) regarding error control.

## 3.4 Sampling Quality of AdjointDPM

To evaluate the effectiveness of the reparameterization in AdjointDPM, we generate images by solving the original ODE (4) and the reparameterized one (10) respectively. We also use other state-of-the-art samplers to synthesize images and compare the sampling qualities (measured by FID). We follow the implementation of DPM in Song et al. (2021) and use the publicly released checkpoints[2] (trained on the CIFAR10 dataset) to generate images in an unconditional manner. We use the *torchdiffeq* package[3] and solve the ODE (4) and (10) via the Adams–Bashforth numerical solver with order 4. We choose a suitable NFE number for ODE solvers so that the DPM can generate content with good quality while not taking too much time. We compare the performance of AdjointDPM (solving the reparameterzied ODEs) to the case of solving the original ones under small NFE regions ($\leq 50$).

---

[2]https://github.com/yang-song/score_sde
[3]https://github.com/rtqichen/torchdiffeq

Table 1: FID ($\downarrow$) for VPSDE models evaluated on CIFAR10 under small NFE regions.

| NFE | Solving (4) | DPM-solver | Solving (10) |
|---|---|---|---|
| 10 | 9.50 | 4.70 | 4.36 |
| 20 | 8.27 | 2.87 | 2.90 |
| 50 | 5.64 | 2.62 | 2.58 |

We generate the same number of images as the training set and compute the FID between the generated images and the real ones. From Table 1, we observe that, after reparameterizing the forward generation process to a non-stiff ODE function, we can generate higher-quality samples with lower FID values under the same NFEs. The sampling qualities of our method are also comparable to those of the state-of-the-art sampler (DPM-solver (Lu et al., 2022a)).

## 4 APPLICATIONS

In this section, we apply AdjointDPM to perform several interesting tasks, involving optimizing initial noisy states or model weights for performing guided sampling or customized generation. Due to the space limitation, we provide another application using AdjointDPM for converting visual effects into identification prompt embeddings in Appendix A. The experimental results of all applications demonstrate that our method can effectively back-propagate the loss information on the generated images to the related variables of DPMs.

### 4.1 GUIDED SAMPLING

In this section, we use AdjointDPM for guided sampling. The guidance is defined by the loss on the output samples, such as the classification score. We aim to optimize the sampling trajectory, $\{\mathbf{x}_t\}_{t=T}^1$, to make the generated images satisfy certain requirements.

### 4.1.1 VOCABULARY EXPANSION

The publicly released Stable Diffusion model is pre-trained on a very large-scale dataset (*e.g.*, LAION (Schuhmann et al., 2022)), where the images are captioned at a high level. It can generate diverse general objects. However, when using it for synthesizing a specific kind of object, such as certain breeds of animals or species of plants, we may obtain suboptimal results in the sense that the generated images may not contain subtle characteristics. For example, when generating a picture of the "Cairn" dog, the Stable Diffusion model can synthesize a dog picture but the shape and the outer coat may mismatch.

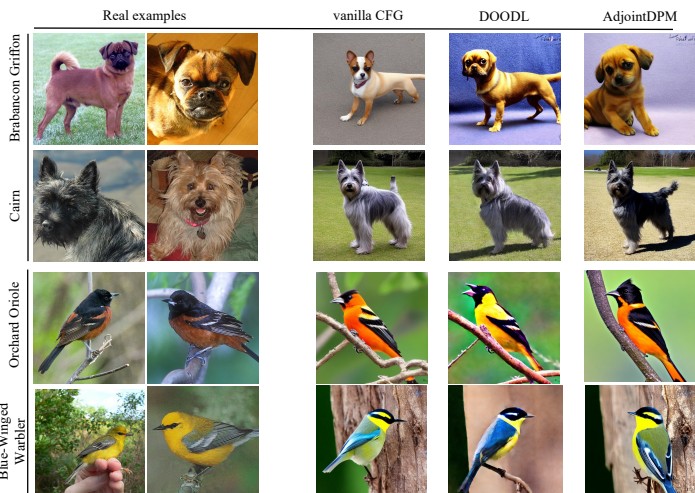

Figure 1: Examples for Vocabulary Expansion. The original Stable Diffusion cannot generate samples whose features exactly match the ground-truth reference images. Using the FGVC model, AdjointDPM can guide the Stable Diffusion to synthesize a certain breed of animals. Here we can generate images where the dog's face closely resembles target breeds. Besides, we generate birds with features that are more similar to real images, such as black heads for Orchard Oriole and blue feathers for the Blue-Winged Warbler.

Here, we use a fine-grained visual classification (FGVC) model as guidance to help the Stable Diffusion model generate specific breeds of dogs. The FGVC model can distinguish objects with subtle variations between classes. Under the guidance of a dog-FGVC model, diffusion models can accurately generate dog images of a specific breed. In other words, *the vocabulary base of the diffusion model gets expanded*. We formulate this task as follows: Let $f(\cdot)$ denote the FGVC model. The guidance $L(y, f(\mathbf{x}_0))$ is defined as the prediction score of the generated image $\mathbf{x}_0$ for class $y$. During sampling, in each time step $t$, we obtain the gradient of guidance $L$ with respect to the noisy state $\mathbf{x}_t$, namely $\frac{\partial L}{\partial \mathbf{x}_t}$, to drive the sampling trajectory.

Table 2: Per-Class % FID change ($\downarrow$).

| Dataset | DOODL | AdjointDPM |
|---------|-------|------------|
| Dogs    | $-5.3\%$ | $-7.1\%$ |

We present the visual and numerical results in Fig. 1 and Table 2, respectively. By visual comparison to the vanilla Stable Diffusion (SD), we observe that under the guidance of AdjointDPM, the color and outer coat of the generated dog images align better with the ground-truth reference pictures. Besides, we compute the reduced FID values on Stanford Dogs (Dogs) (Khosla et al., 2011) and find that the FID values are also improved. We also compare AdjointDPM to a state-of-the-art baseline, DOODL (Wallace et al., 2023a). AdjointDPM outperforms in terms of visual quality and reduced FID values compared to SD. Refer to more results, optimization details, and comparison with the existing models in the Appendix B.

### 4.1.2 SECURITY AUDITING

DPMs like Stable Diffusion have been widely used in content creation platforms. The large-scale datasets used for training may contain unexpected and harmful data (*e.g.*, violence and pornography). To avoid generating NSFW content, AI generation systems are usually equipped with a safety filter that blocks the outputs of potentially harmful content. However, deep neural networks have been shown to be vulnerable against adversarial examples (Goodfellow et al., 2015; Yan et al., 2020). *This naturally raises the concern—may existing DPM generation systems output harmful content that can sneakily bypass the NSFW filter?* Formally, denote $f(\cdot)$ as the content moderation filter and $c$ as the conditioning prompts containing harmful concepts. We randomly sample an initial noisy state $\mathbf{x}_T$, the generated image $\Phi(\mathbf{x}_T, c, \epsilon_\theta)$ will likely be filtered out by $f(\cdot)$. We want to audit the security of generation systems by searching for another initial noisy state $\mathbf{x}'_T$, which lies in a small $\delta$-neighborhood of $\mathbf{x}_T$, such that the corresponding output $\Phi(\mathbf{x}'_T, c, \epsilon_\theta)$ may still contain harmful content but bypass $f(\cdot)$. If we find it, the generation systems may face a serious security issue.

This problem can also be formulated as a guided sampling process. The guidance $L$ is defined as the distance between harmful prompt $c$ and the prediction score $f(\Phi(\mathbf{x}'_T, c, \epsilon_\theta))$. The distance is measured by the similarity between CLIP embeddings. We optimize the perturbation $\delta$ on $\mathbf{x}_T$ to maximize the distance. The norm of the perturbation is limited to be $\tau$ as we want to ensure the newly generated image is visually similar to the original one.

$$\max_{\delta: \|\delta\|_\infty \leq \tau} L(c, f(\Phi(\mathbf{x}_T + \delta, c, \epsilon_\theta))).$$

We use AdjointDPM to solve this optimization problem and find that there indeed exist initial noisy states of the Stable Diffusion model, whose corresponding outputs can mislead the NSFW filter. Results are shown in Fig. 2. This observation raises an alert about the security issues of AI generation systems. *Our research community has to develop more advanced mechanisms to ensure the isolation between NSFW content and users, especially teenagers.*

Table 3: Success ratio (%) of adversarial initial states bypass the classifier. We show the results for five classes from the ImageNet dataset.

| Index | 242 | 430 | 779 | 859 | 895 |
|-------|-----|-----|-----|-----|-----|
| Ratio | 63.9 | 75.8 | 45.3 | 58.22 | 52.6 |

For numerical evaluation, we also audit the security of a smaller diffusion model which is trained on the ImageNet dataset. We chose ten classes and sampled hundreds of images. For each sample, we search for the adversarial perturbation that maximizes the classification error. We record the resultant noisy states that can mislead the classification module. The success ratio achieves around *51.2%*. Qualitative results about ImageNet are shown in Fig. 7 and more experimental details are provided in Appendix C.

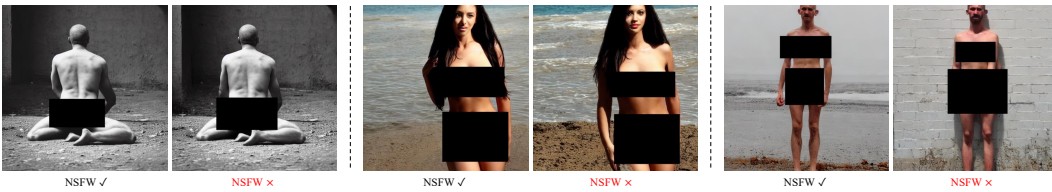

Figure 2: Adversarial samples against the NSFW filter. We show the image generated by conditioning on harmful prompts (*e.g.*, *"A photograph of a naked man"*) on the left. These images will be blocked by the NSFW filter. However, the images generated from adversarial initial noises circumvent the NSFW filter (Black squares are added by authors for publication).

## 4.2    STYLIZATION VIA A SINGLE REFERENCE

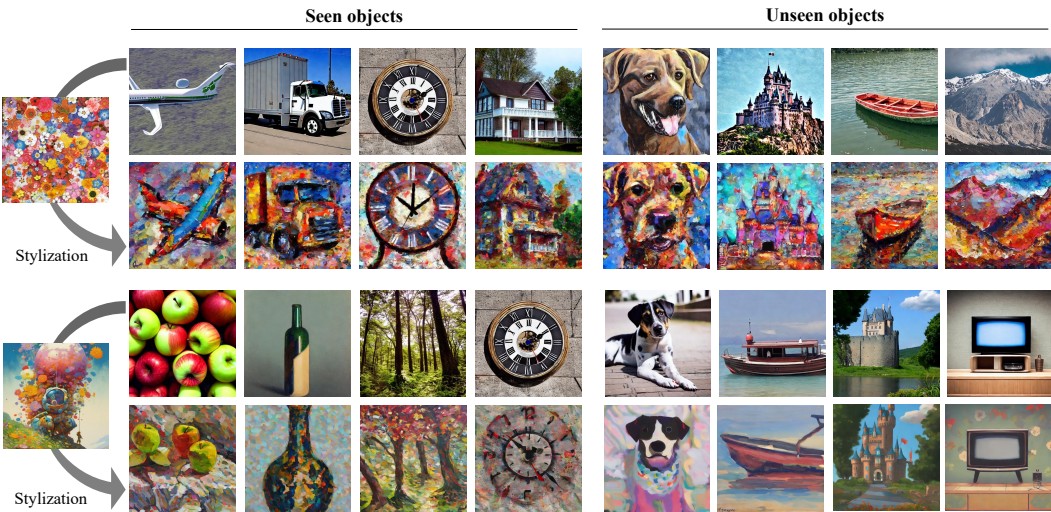

Figure 3: Stylization examples. Images generated by the original Stable Diffusion are shown at the top. The bottom are samples of the stylized Stable Diffusion.

We consider using AdjointDPM to fine-tune the weights of the UNet in Stable Diffusion for stylization based on a single reference image. We define the style of an image $\mathbf{x}$ by its Gram matrix (Gatys et al., 2015), which is computed based on the features extracted from a pre-trained VGG model.[4] Here we denote the features extracted from the VGG as $\mathbf{F}(\mathbf{x})$ and the Gram matrix $\mathbf{G}(\mathbf{x}) = \mathbf{F}\mathbf{F}^T$.

Given a reference image, we denote the target style as $\mathbf{G}_{\text{style}}$. We aim to fine-tune the weights of the UNet so that the style of generated images matches the target one. This task is formulated as the optimization problem (13). The objective contains two terms. Besides the style loss $L_{\text{s}}$ (mean squared error), we also add a term of content loss $L_{\text{c}}$. The content loss encourages the model to preserve the ability to generate diverse objects when adapting itself to a certain style. In specific, we sample multiple noise-prompt-image triplets, $\{(\mathbf{x}_T^i, c^i, \mathbf{x}_0^i)\}_{i=1}^N$, where $\mathbf{x}_0^i$ denotes the clean image generated by the pre-trained Stable Diffusion with the input of $(\mathbf{x}_T^i, c^i)$. The content loss is defined as the mean squared error between the features of originally generated images $\mathbf{F}(\mathbf{x}_0^i)$ and those generated by optimized weights $\mathbf{F}(\Phi(\mathbf{x}_T^i, c^i, \epsilon_\theta))$. Two coefficients, $w_{\text{s}}$ and $w_{\text{c}}$, balance the strength of the two loss terms.

$$\min_\theta \frac{1}{N} \sum_i \left[ w_{\text{s}} L_{\text{s}} \left( \mathbf{G}_{\text{style}}, \mathbf{G}(\Phi(\mathbf{x}_T^i, c^i, \epsilon_\theta)) \right) + w_{\text{c}} L_{\text{c}} \left( \mathbf{F}(\mathbf{x}^i), \mathbf{F}(\Phi(\mathbf{x}_T^i, c^i, \epsilon_\theta)) \right) \right] \tag{13}$$

---

[4]https://pytorch.org/tutorials/advanced/neural_style_tutorial.html

Table 4: CLIP similarity scores (↑) between samples and the conditioning prompts.

| Prompt | DreamBooth | Text-Inv | AdjointDPM |
|--------|-----------|----------|------------|
| Airplane | 21.98 | 24.44 | 27.34 |
| Clock | 28.23 | 26.74 | 30.00 |
| House | 25.40 | 25.91 | 29.03 |
| Cat | 25.90 | 22.30 | 26.83 |
| Apples | 28.10 | 24.06 | 28.59 |

We construct 10 prompts corresponding to ten of the CIFAR100 classes and sample starting noises to generate 100 images (10 images for each prompt) to form our training dataset. Visual and numerical results are shown in Fig. 3 and Table 4 respectively. We observe the SD model fine-tuned by AdjointDPM can generate stylized images of different objects. The stylizing capability also generalizes to the concepts unseen during fine-tuning (shown in the right part of Fig. 3). In addition to the high visual quality, the samples also align well with the conditioning prompts according to the high CLIP similarity scores. We compare AdjointDPM with other methods for stylization, including DreamBooth (Ruiz et al., 2022) and Textural-Inversion (Gal et al., 2022) (see the qualitative comparisons in Fig. 12). We observe that AdjointDPM achieves better alignment between image samples and the prompts. In addition, these existing methods barely can be generalized to unseen objects in this case only one reference image is available. More details and examples of stylization are shown in Appendix D.

## 5 RELATED WORKS AND DISCUSSION

**Customization of Text-to-Image Generation** Text-to-image customization aims to personalize a generative model for synthesizing new images of a specific target property. Existing customization methods typically tackle this task by either representing the property via a text embedding (Gal et al., 2022; Mokady et al., 2022; Wen et al., 2023) or finetuning the weights of the generative model (Ruiz et al., 2022; Kawar et al., 2023; Han et al., 2023). For example, Textual-Inversion (Gal et al., 2022) inverts the common identity shared by several images into a unique textual embedding. To make the learned embedding more expressive, Daras & Dimakis (2022) and Voynov et al. (2023) generalize the unique embedding to depend on the diffusion time or the layer index of the denoising UNet, respectively. In the other line, DreamBooth (Ruiz et al., 2022) learns a unique identifier placeholder and finetunes the whole diffusion model for identity customization. To speed up and alleviate the overfitting, Custom Diffusion (Kumari et al., 2022) and SVDiff (Han et al., 2023) only update a small subset of weights. Most of these existing methods assume that a handful of image examples (at least 3-5 examples) sharing the same concept or property are provided by the user in the first place. Otherwise, the generalization of resultant customized models usually will be degraded, *i.e.*, they barely can synthesize unseen objects (unseen in the training examples) of the target concept. However, in some cases, it is difficult or even not possible to collect enough data that can represent abstract requirements imposed on the generated content. For example, we want to distill the editing effects/operations shown in a single image by a media professional or a novel painting style of a unique picture. In contrast, this paper relaxes the requirement of data samples and proposes the AdjointDPM for model customization only under the supervision of a differentiable loss.

**Guidance of Text-to-Image Generation** Concerning the guidance of diffusion models, some algorithms (Bansal et al., 2023; Yu et al., 2023) mainly use the estimated clean state for computing the guidance loss and the gradient with respect to intermediate noisy states. The gap to the actual gradient is not negligible. As a result, the guided synthesized results may suffer from degraded image quality. Instead, some other methods (Wallace et al., 2023a; Liu et al., 2023b), such as DOODL, compute the gradients by exploiting the invertibility of diffusion solvers (*e.g.*, DDIM). This paper also formulates the sampling process as ODE and proposes AdjointDPM to compute the gradients in a more accurate and adaptive way.

The main contribution of this paper is that we propose a ubiquitous framework to optimize the related variables (including UNet weights, text prompts, and latent noises) of diffusion models based on the supervision information (any arbitrary differentiable metric) on the final generated data. To our best knowledge, AdjointDPM is the first method that can compute the gradients for all types of parameters of diffusion models. In contrast, other methods, such as DOODL (Wallace et al., 2023a), FlowGrad (Liu et al., 2023b), and DEQ-DDIM (Pokle et al., 2022)), either only work for the noisy states or require the diffusion sampling process having equilibrium points. In the future, we will explore more real-world applications of AdjointDPM.

ACKNOWLEDGMENTS

This research/project is supported by the Singapore Ministry of Education Academic Research Fund (AcRF) Tier 2 under grant number A-8000423-00-00 and the Singapore Ministry of Education AcRF Tier 1 under grant number A-8000189-01-00. We would like to acknowledge that the computational work involved in this research work is partially supported by NUS IT's Research Computing group.

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

CONTENTS

## A   TEXT EMBEDDING INVERSION

In addition to the applications shown in Section 4, we consider another application concerning using AdjointDPM to convert visual effects (*e.g.*, bokeh and relighting) into an identification text embedding #. Suppose we are given an image pair, namely an original image and its enhanced, the enhanced version is edited by some professional and appears with certain fascinating visual effects. After optimization, we can combine the obtained embedding with various text prompts to generate images with the same visual effect. Here, we simulate this real setting by using a text-to-image model to generate images with and without a certain effect.

Suppose we are provided a text-to-image DPM $\Phi(\cdot)$, we can generate an image $\mathbf{x}$ by denoising randomly sampled noise $\mathbf{x}_T$ in the condition of the base prompt $c_{\text{base}}$. We further improve the aesthetic quality by inserting some keywords $c_{\text{target}}$, like "bokeh", into the conditioning prompt. The newly generated images are denoted by $\mathbf{x}^*$. We use $\mathbf{x}^*$ or its feature as a reference to define a loss $L(\cdot)$ that measures the distance to the target effect, such as the $\ell_2$ or perceptual loss. We aim to optimize a special embedding # that can recover the visual effects in $\mathbf{x}^*$:

$$\min_{\#} L\left(\mathbf{x}^*, \Phi(\mathbf{x}_T, \{c_{\text{base}}, \#\}, \epsilon_\theta)\right).$$

We utilize the publicly released Stable Diffusion models for image generation and set the loss function as the mean squared error (MSE) between the target images and the generated images. We aim to optimize a prompt embedding in the CLIP (Radford et al., 2021) embedding space and use the obtained embedding for image generation by concatenating it with the embeddings of other text prompts. As shown in Fig. 4, we observe AdjointDPM successfully yields an embedding # that can ensure the appearance of target visual effects, including the *bokeh* and *relighting*. Furthermore, the obtained embeddings # also generalize well to other starting noise and other text prompts. For example, the bokeh-# is optimized on a pair of *totoro* images; it also can be used for generating different images of *totoro* and other objects like *dog*. Similarly, the obtained # corresponding to manual editing ("converting to black and white") also can be used for novel scene generation.

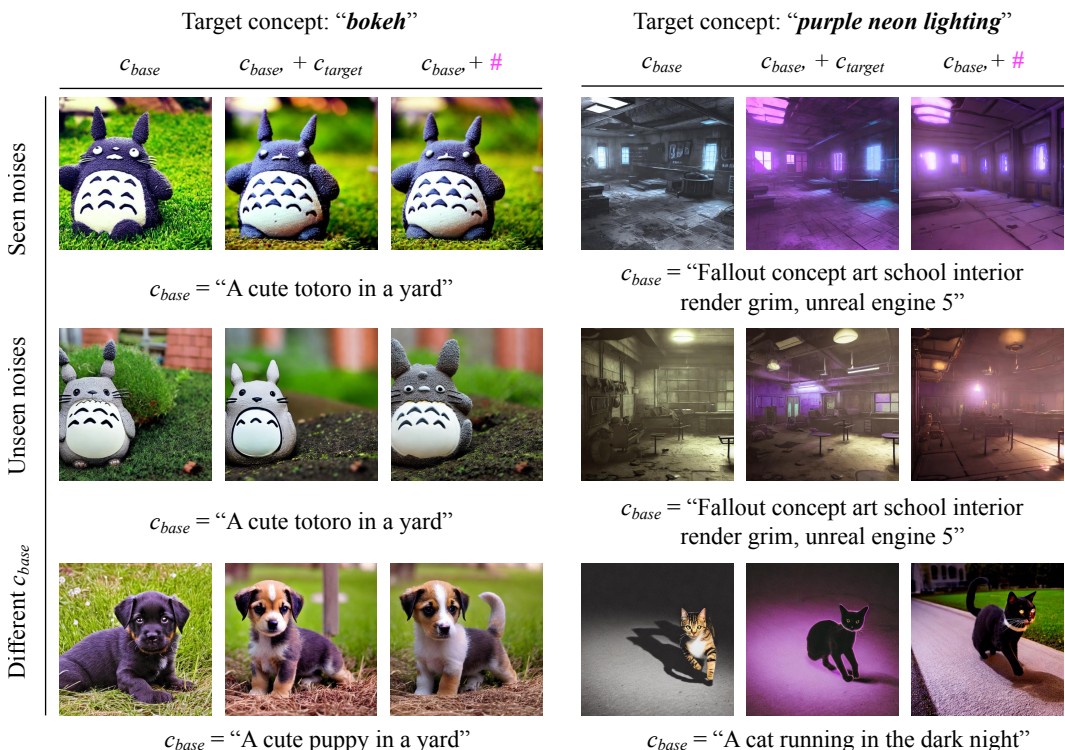

Figure 4: Examples on prompt inversion - part 1

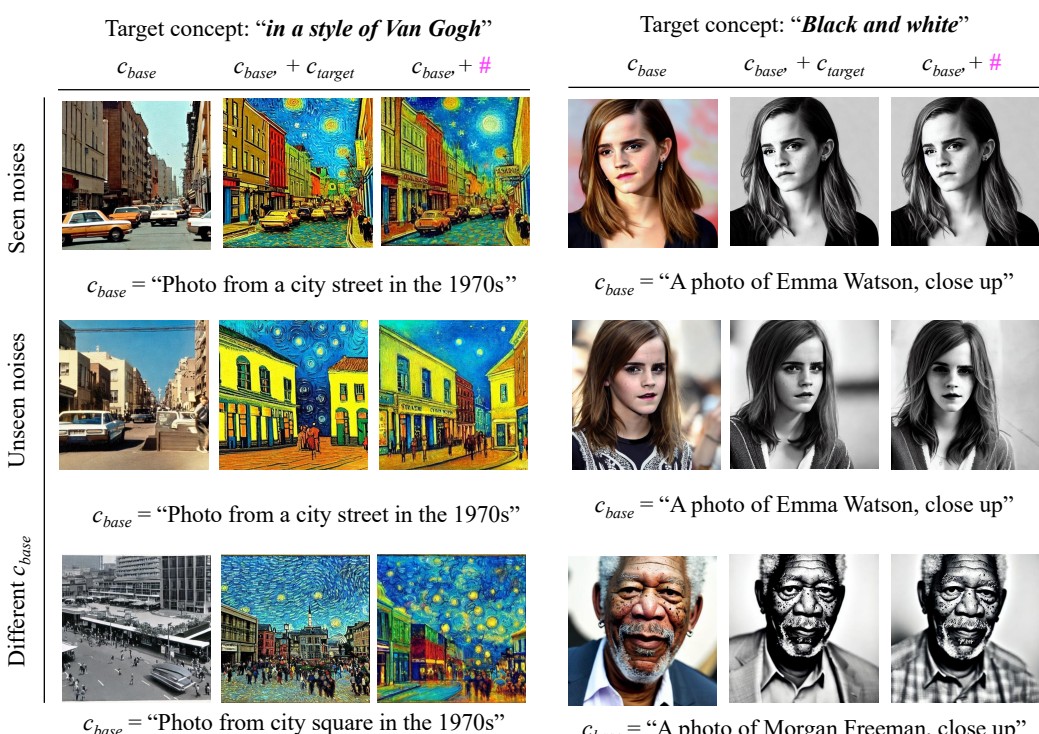

Figure 5: Examples on prompt inversion - part 2

## B  EXPERIMENTAL DETAILS ON VOCABULARY EXPANSION AND COMPARISON WITH EXISTING MODELS

**Comparison with Existing Models.**  DOODL (Wallace et al., 2023a) optimizes the initial diffusion noise vectors w.r.t a model-based loss on images generated from the full-chain diffusion process. In their work, they obtain the gradients of loss w.r.t noise vectors by using invertible neural networks (INNs). There are three main differences between DOODL and our work. First, while DOODL optimizes the initial diffusion noise vectors, our work optimizes related variables, including network parameters, initial noises and textual embeddings w.r.t a model-based loss on images generated from the full-chain diffusion process. Thus, we consider the broader cases of DOODL. Second, in the calculation of gradients w.r.t initial noises, DOODL uses the invertibility of EDICT (Wallace et al., 2023b), i.e., $x_0$ and $x_T$ are invertible. This method does not apply to the calculation of gradients w.r.t. network parameters and textual embeddings as they share across the full-chain diffusion process. Finally, with regard to the memory consumption when calculating gradients with respect to the initial noise, our experimental results are as follows: we utilized the stable diffusion v1.5 checkpoint to run both the AdjointDPM and DOODL models on a V100 GPU (32GB memory). For the AdjointDPM method, backpropagating the gradients with respect to a single initial noise required 19.63GB of memory. In comparison, the DOODL method consumed 23.24GB for the same operation. The additional memory consumption in DOODL is mainly from the dual diffusion process in EDICT. Thus, our method is more efficient in memory consumption. In terms of time consumption, DOODL relies on the invertibility of EDICT, resulting in identical computation steps for both the backward gradient calculation and the forward sampling process. Besides, they usually use DDIM sampling methods, which is equivalent to the first-order neural ODE solver. However, our AdjointDPM methods have the flexibility to apply high-order ODE solvers, allowing for faster backward gradient calculation. See the following for an experimental comparison between our method and DOODL.

We also make a comparison with FlowGrad (Liu et al., 2023b). FlowGrad efficiently backpropagates the output to any intermediate time steps on the ODE trajectory, by decomposing the backpropagation and computing vector Jacobian products. FlowGrad focuses on refining the ODE generation paths to the desired direction. This is different from our work, which focuses on the finetuning of related variables, including network parameters, textual embedding and initial noises of diffusion models for customization. Besides, FlowGrad methods also can not obtain the gradients of loss w.r.t. textual embeddings and neural variables as these variables share across the whole generation path. Then for the gradients w.r.t the latent variables, we could show the memory consumption of our methods is constant while they need to store the intermediate results.

**Experimental Details on Vocabulary Expansion**  During the optimization of the noise states under the guidance of FGVC model, we adopt the Euler ODE solver in our AdjointDPM method with 31 steps. We optimize the noise states using the AdamW optimizer for 30 epochs with different learning rates for different breeds. For the implementation of DOODL, we follow the officially released code[5] and we set the sampling steps also to be 31 and optimization steps for 30. Following the DOODL, we also measure the performance by computing the FID between a set of generated images (4 seeds) and the validation set of the FGVC dataset being studied. We do experiments on Stanford Dogs (Dogs) (Khosla et al., 2011) datasets and calculate FID values. More qualitative results are shown in Fig 6.

## C  EXPERIMENTAL DETAILS AND MORE RESULTS ON SECURITY AUDITING

In this section, we provide explicit details about generating adversarial examples against an ImageNet classifier and the NSFW filter in Stable Diffusion, respectively.

**Security Auditing under an ImageNet Classifier.**  To generate adversarial samples against an ImageNet classifier, we follow the implementation of classifier guidance generation of DPM[6] and use

---

[5] https://github.com/salesforce/DOODL
[6] https://github.com/LuChengTHU/dpm-solver/tree/main/examples/ddpm_and_guided-diffusion

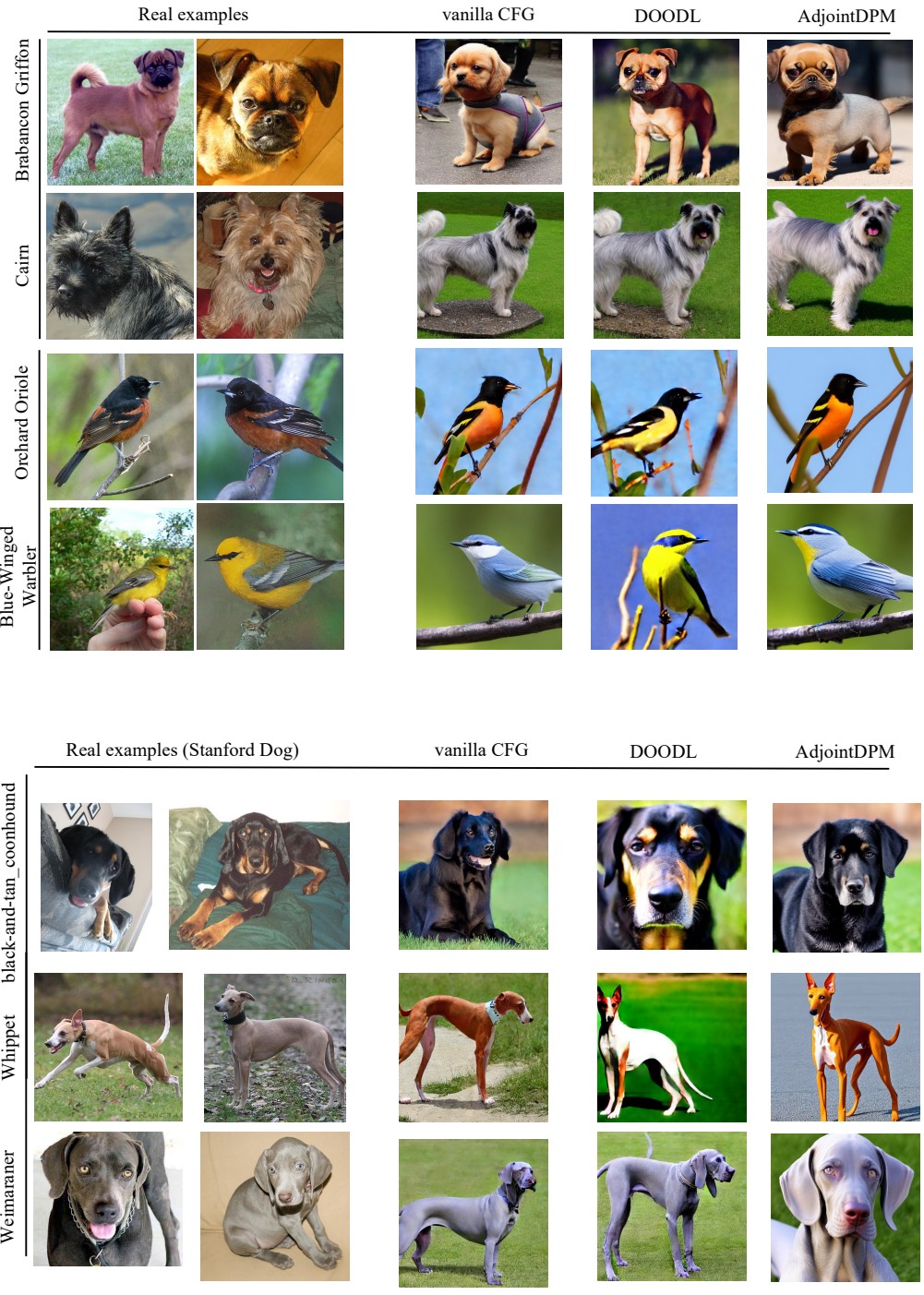

Figure 6: More Examples for Vocabulary Expansion.

the publicly released checkpoints trained on the ImageNet 128x128 dataset to generate images in a conditional manner. We adopt the pre-trained ResNet50[7] as our ImageNet classifier.

To generate adversarial examples, we first randomly choose an ImageNet class and set it as the class label for classifier guidance generation, and then we pass the generated images to the ResNet50 classifier. If the outputs of ResNet50 classifier are aligned with the chosen class label, we begin to do an adversarial attack by using AdjointDPM. For the adversarial attack, we adopt the targeted attack, where we choose a target class and make the outputs of ResNet50 close to the pre-chosen target class by minimizing the cross entropy loss. We also clamp the updated initial noise in the range of $[x_T - 0.8, x_T + 0.8]$ (i.e., set $\tau = 0.8$ in Sec. 4.3 to ensure that generated images do not visually change too much compared with the start images). We show more adversarial examples against the ImageNet classifier in Fig. 7. Besides, define the attack rate as the ratio between the number of samples with incorrect classification results after the attack and the total number of samples. We also get the attack rate 51.2% by generating 830 samples from 10 randomly chosen classes. The class labels here we choose are $[879, 954, 430, 130, 144, 242, 760, 779, 859, 997]$.

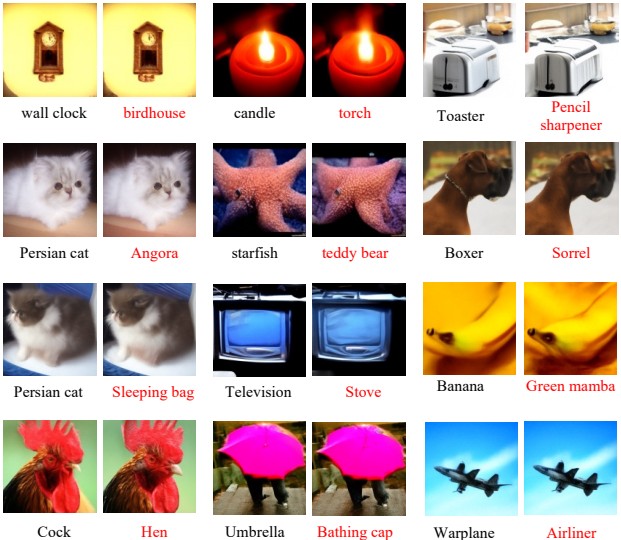

Figure 7: Adversarial examples against the ImageNet classifier. We show the originally generated images with their class names on the left; these images are correctly classified by ResNet50. On the right, we show the corresponding adversarial images which successfully mislead the classifier.

**Experimental Details on the NSFW Filter.** In this case, we set $\tau = 0.9$. We follow the implementation of Stable Diffusion[8] and set the loss function as the cosine distance between CLIP embeddings Radford et al. (2021) of generated images and unsafe embeddings from Rando et al. (2022).

## D  MORE EXAMPLES ON FINETUNING WEIGHTS FOR STYLIZATION

In this section, we introduce the experimental details of stylization and present more stylized examples on seen noises and seen classes, unseen noises and seen classes, and unseen noises and unseen classes.

For training, we choose ten classes from CIFAR-100 classes, which are ["An airplane", "A cat", "A truck", "A forest", "A house", "sunflowers", "A bottle", "A bed", "Apples", "A clock"]. Then we randomly generate 10 samples from each class to compose our training dataset. Besides, we directly use these class names as the input prompt to Stable Diffusion [9]. We optimize the parameters of cross

---

[7]https://pytorch.org/vision/stable/models.html
[8]https://github.com/huggingface/diffusers
[9]https://github.com/CompVis/stable-diffusion

attention layers of UNet for 8 epochs by using AdamW optimizer with learning rate $10^{-4}$. We show more stylization results on 100 training samples (seen noises and seen classes) in Fig. 8. Meanwhile, we also show more examples of seen-classe-unseen-noise and examples of unseen-class-unseen-noise in Fig. 10. In Fig. 11, we also show the stylization results on other target style images, in which one is downloaded from the showcase set of Midjourney[10] and the other is the Starry Night by Van Gogh. We also present that the finetuned networks under ODE forms can still apply SDE solvers (such as DDPM) in Fig. 9.

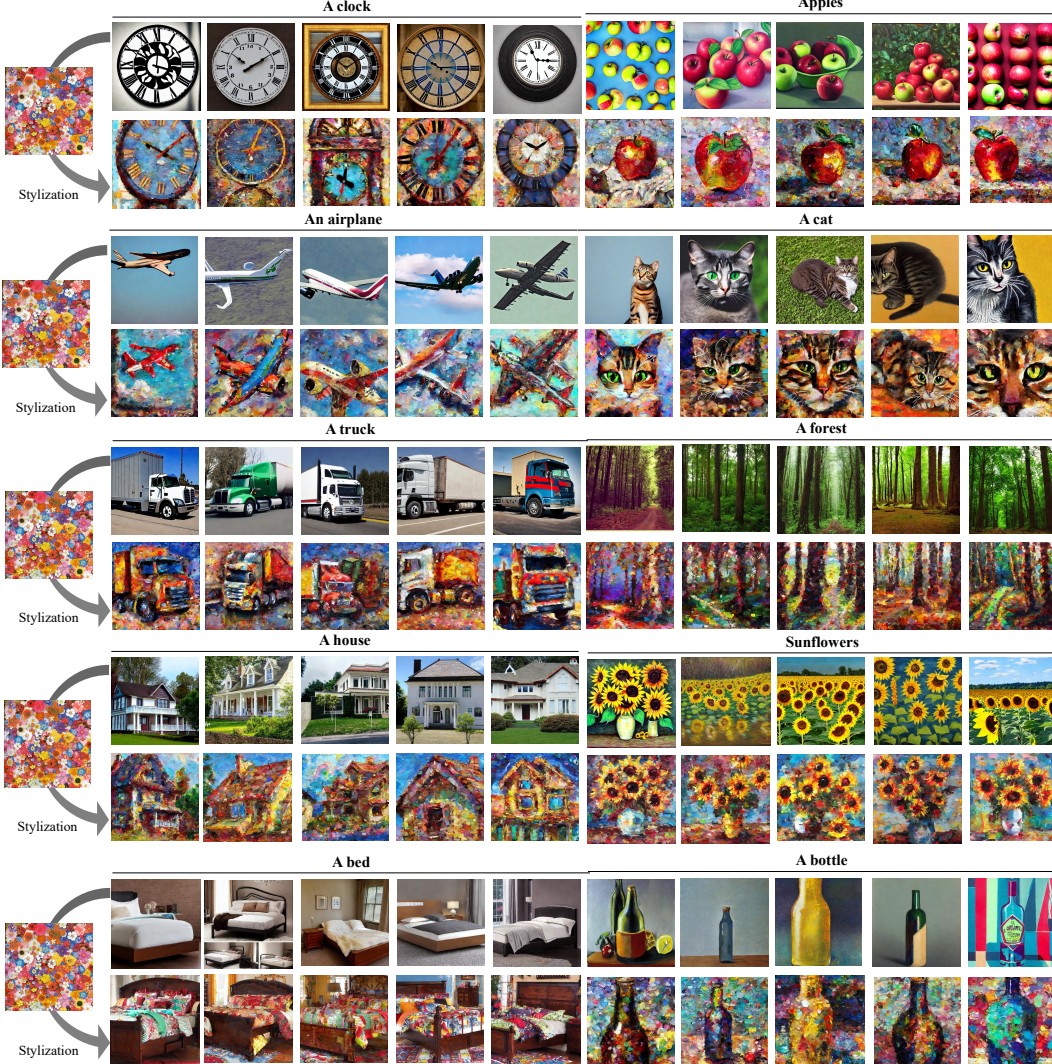

Figure 8: Stylization examples on seen classes and seen noises

## D.1 QUALITATIVE COMPARISONS TO TEXTUAL-INVERSION AND DREAMBOOTH

We also provide visual comparisons to Textual Inversion Gal et al. (2022) and DreamBooth Ruiz et al. (2022) in Fig. 12. We follow the implementation of Textual Inversion[11] and DreamBooth[12]. For textual inversion, we use the same target style image in Section 4.2 as the training dataset. As in our AdjointDPM model, we use one style image for training, for fair comparison, the training

---

[10] https://cdn.midjourney.com/61b8bd5d-846b-4f69-bdc1-0ae2a2abcce8/grid_0.webp

[11] https://huggingface.co/docs/diffusers/training/text_inversion

[12] https://huggingface.co/docs/diffusers/training/dreambooth

# Seen class unseen noise

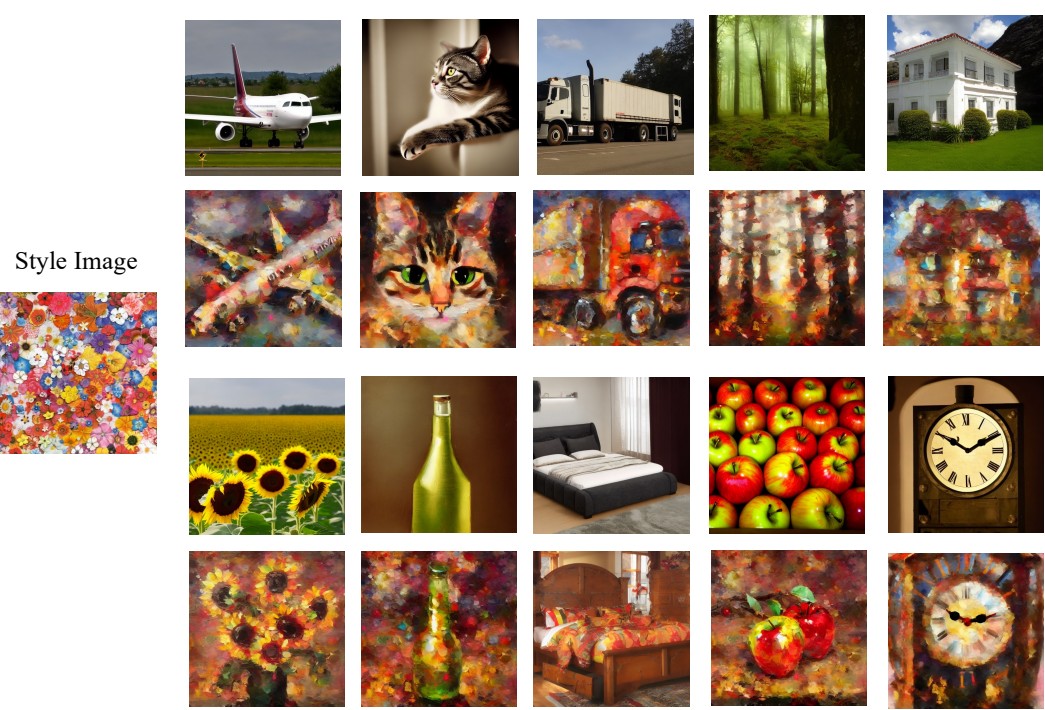

Style Image

Figure 9: Stylization examples on seen classes and unseen noises using DDPM.

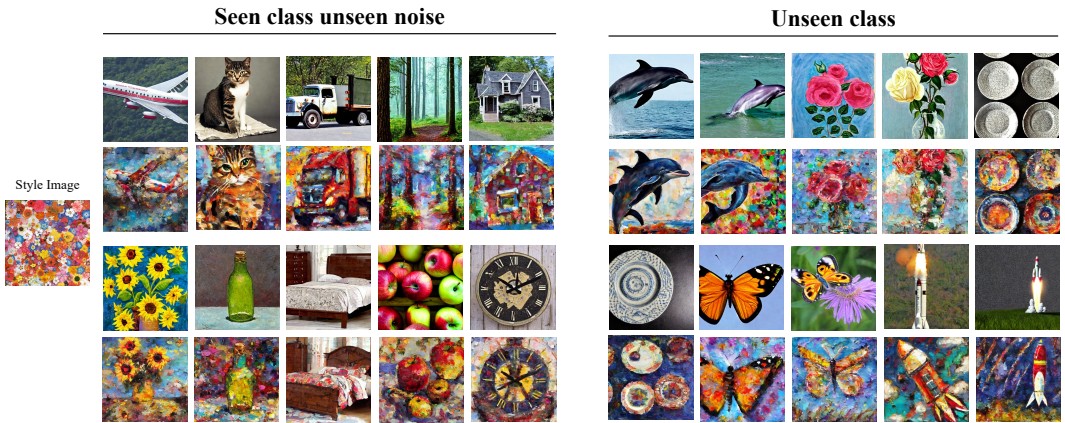

Figure 10: Stylization examples on *seen*-class-*unseen*-noise and *unseen*-class

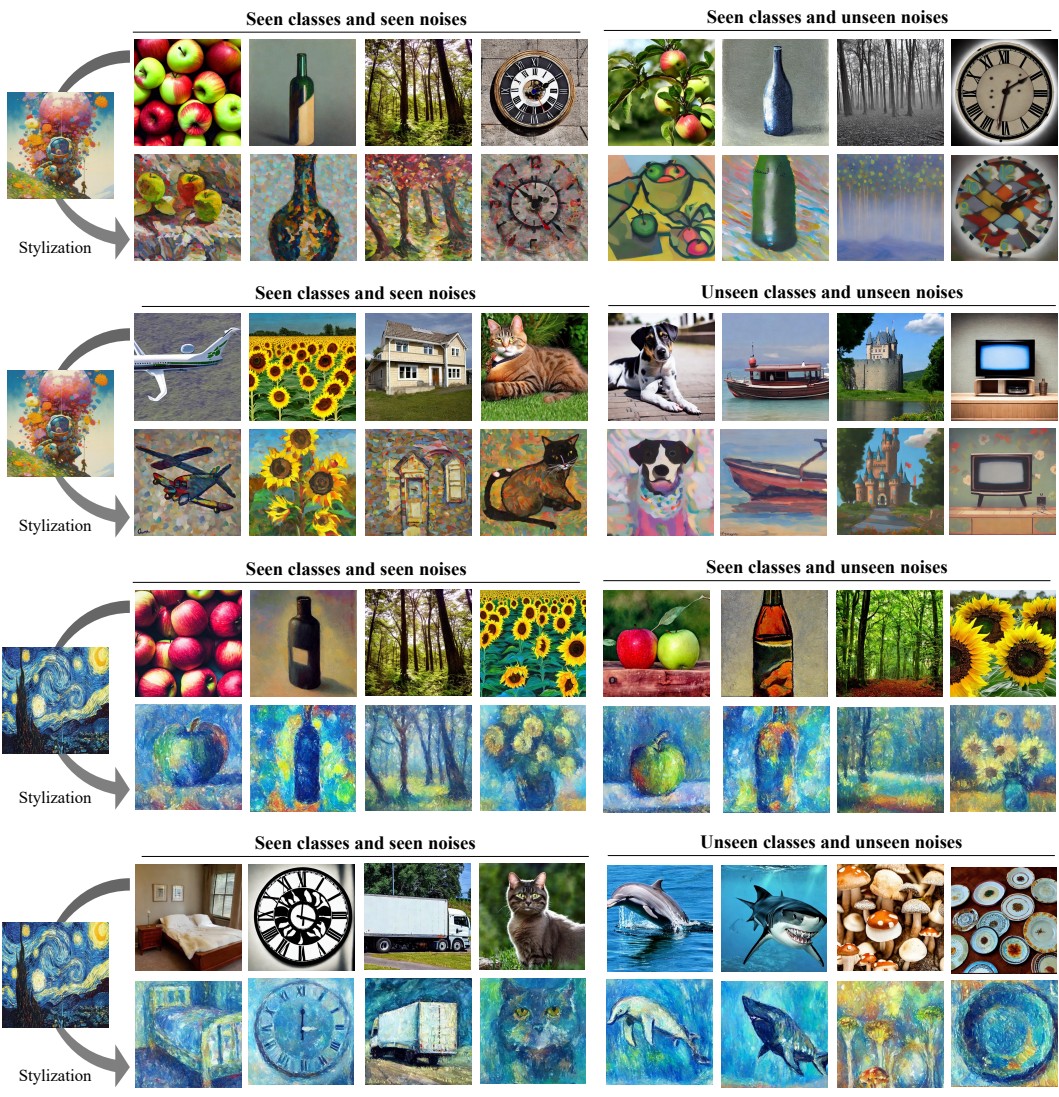

Figure 11: Stylization examples on other style images

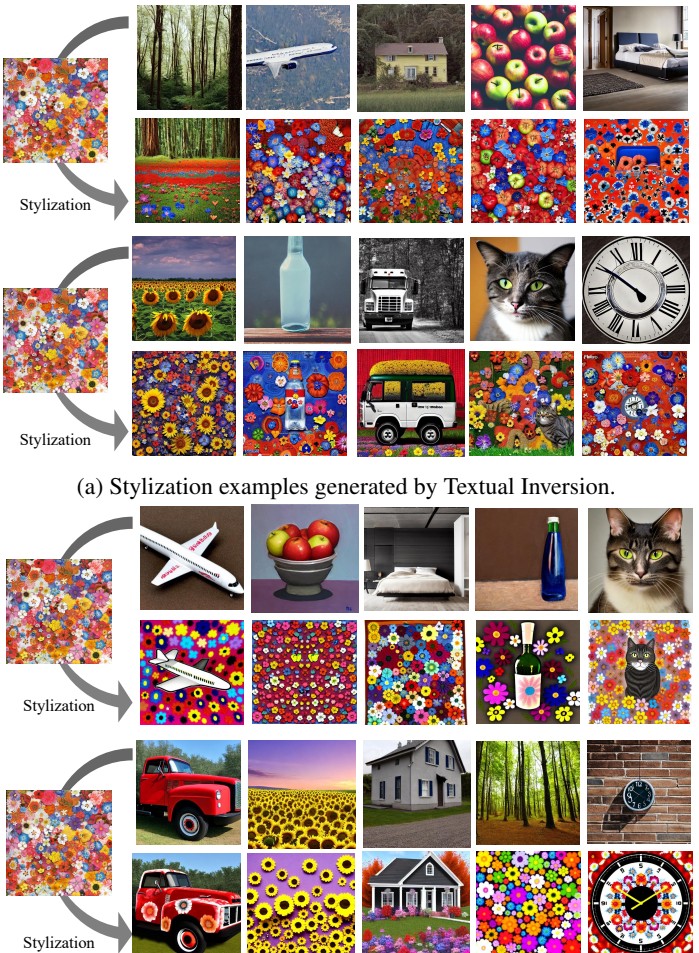

(a) Stylization examples generated by Textual Inversion.

(b) Stylization examples generated by DreamBooth.

Figure 12: Comparison to Textual-Inversion and DreamBooth for stylization.

dataset to Textual Inversion and DreamBooth also include only one style image. We set the *learnable property* as "style", *placeholder token* as "<bengiles>", *initializer token* as "flowers" in Textual Inversion. Then we run 5000 epochs with a learning rate $5 \times 10^{-4}$ to train the Textual Inversion. For DreamBooth, we set *instance prompt* as "bengiles flowers". Then we run 1000 epochs with a learning rate $1 \times 10^{-6}$ to train the DreamBooth. In Fig. 12, we show the stylization examples generated by using Textual Inversion and DreamBooth. We can observe distinct differences in the stylization outcomes when comparing the Textual Inversion and DreamBooth approaches to our AdjointDPM methods. In some cases, for Textual Inversion and DreamBooth, we have noticed that the main objects within an image can vanish, resulting in the entire image being predominantly occupied by the applied "style."

## E    IMPLEMENTATION OF ADJOINTDPM

In this section, we present the explicit AdjointDPM algorithm. For VP-SDE, we have $f(t) = \frac{\mathrm{d} \log \alpha}{\mathrm{d}t}$ and $g^2(t) = \frac{\mathrm{d}\sigma_t^2}{\mathrm{d}t} - 2\frac{\mathrm{d} \log \alpha}{\mathrm{d}t}\sigma_t^2$. Based on the definition of $\mathbf{y}_t$ and $\rho$, we obtain

$$\mathbf{y}_t = \frac{\alpha_0}{\alpha_t}\mathbf{x}_t, \quad \rho = \gamma(t) = \alpha_0 \frac{\sigma_t}{\alpha_t} - \sigma_0.$$

We denote the timesteps for solving forward generation ODE as $\{t_i\}_{i=0}^N$, where $N$ is the number of timesteps. Then based on this re-parameterization, we can show our explicit forward generation algorithm and reverse algorithm of obtaining gradients for VP-SDE in Algorithm 2 and Algorithm 3.

For the choice of $\alpha_t$, $\sigma_t$, and sampling steps $\{t_i\}_{i=1}^N$, we adopt the implementation of DPM-solver[13]. Specifically, we consider three options for the schedule of $\alpha_t$ and $\sigma_t$, discrete, linear, and cosine. The detailed formulas for obtaining $\alpha_t$ and $\sigma_t$ for each schedule choice are provided in (Lu et al., 2022a, Appendix D.4). The choice of schedule depends on the specific applications. We usually solve the forward generation ODE function from time $T$ to time $\epsilon$ ($\epsilon > 0$ is a hyperparameter near 0). Regarding the selection of discrete timesteps $\{t_i\}_{i=1}^N$ in numerically solving ODEs, we generally divide the time range $[T, \epsilon]$ using one of three approaches: uniform, logSNR, or quadratic. The specific time splitting methods can be found in the DPM-solver. Subsequently, we obtain the generated images and gradients by following Algorithm 2 and Algorithm 3. To solve ODE functions in these algorithms, we directly employ the *odeint adjoint* function in the *torchdiffeq* packages[14].

---

**Algorithm 2** Forward generation by solving an ODE initial value problem

---

**Input:** model $\boldsymbol{\epsilon}_\theta$, timesteps $\{t_i\}_{i=0}^N$, initial value $\mathbf{x}_{t_0}$.

$\quad \mathbf{y}_{t_0} \leftarrow \mathbf{x}_{t_0};$                                                    ▷ Re-parameterize $\mathbf{x}_{t_0}$
$\quad \{\rho_i\}_{i=1}^N \leftarrow \{\gamma(t_i)\}_{i=1}^N;$                                             ▷ Re-parameterize timesteps
$\quad \mathbf{y}_{t_N} = \mathrm{ODESolve}\left(\mathbf{y}_{t_0}, \{\rho_i\}_{i=1}^n, \boldsymbol{\epsilon}_\theta\left(\frac{\alpha_t}{\alpha_{t_0}}\mathbf{y}_t, \gamma^{-1}(t), c\right)\right)$         ▷ Solve forward generation ODE

**Return:** $\mathbf{x}_{t_N} = \frac{\alpha_{t_N}}{\alpha_{t_0}}\mathbf{y}_{t_N}$

---

---

**Algorithm 3** Reverse-mode derivative of an ODE initial value problem

---

**Input:** model $\boldsymbol{\epsilon}_\theta$, timesteps $\{\rho_i\}_{i=1}^N$, final state $\mathbf{y}_{\rho_N}$, loss gradient $\partial L/\partial \mathbf{y}_{\rho_N}$.

$\quad a(\rho_N) = \frac{\partial L}{\partial \mathbf{y}_{\rho_N}}, a_\theta(\rho_N) = \mathbf{0}, z_0 = [\mathbf{y}_{\rho_N}, a(\rho_N), a_\theta(\rho_N)]$         ▷ Define initial augmented state.
$\quad$ **def** AugDynamics($[\mathbf{y}_\rho, \mathbf{a}_\rho, \cdot], \rho, \theta$)                     ▷ Define dynamics on augmented state.
$\qquad$ **return** $[\mathbf{s}(\mathbf{y}_\rho, \rho, \theta, c), -\mathbf{a}_\rho^T \frac{\partial \mathbf{s}}{\partial \mathbf{y}}, -\mathbf{a}_\rho^T \frac{\partial \mathbf{s}}{\partial \theta}]$         ▷ Concatenate time-derivatives
$\quad [\mathbf{y}_{\rho_0}, \frac{\partial L}{\partial \mathbf{y}_{\rho_0}}, \frac{\partial L}{\partial \theta}] = \mathrm{ODESolve}(z_0, \mathrm{AugDynamics}, \{\rho_i\}_{i=1}^N, \theta)$         ▷ Solve reverse-time ODE

**Return:** $[\frac{\partial L}{\partial \mathbf{x}_{t_0}}, \frac{\partial L}{\partial \theta}]$         ▷ Return gradients

---

---

[13] https://github.com/LuChengTHU/dpm-solver
[14] https://github.com/rtqichen/torchdiffeq

