# OpenReview forum: "AdjointDPM: Adjoint Sensitivity Method for Gradient Backpropagation of Diffusion Probabilistic Models"
_ICLR.cc/2024/Conference — ICLR 2024 poster_

### Official Review · Reviewer_CqXY · 2023-10-27

**Soundness:** 3 good
**Presentation:** 3 good
**Contribution:** 3 good
**Rating:** 6
**Confidence:** 3

**Summary:**

This paper...
- proposes AdjointDPM for differentiating through the diffusion sampling process,
- reparametrizes PF ODE and augmented ODE to reduce numerical errors,
- applies AdjointDPM to a wide variety of tasks, such as vocabulary expansion, security auditing, stylization, etc.

**Strengths:**

- The paper is well-written and easy to follow.
- This paper is a nice application of adjoint sensitivity methods to diffusion models. To the best of my knowledge, such application of adjoint sensitivity methods has not been explored before.
- The proposed method can potentially be applied to a wide variety of downstream tasks for diffusion models.

**Weaknesses:**

I must clarify that I am not very familiar with application of diffusion models to tasks such as vocabulary expansion, security auditing, etc. Hence, I am not sure whether the authors have chosen an appropriate and comprehensive set of baselines, or have followed proper evaluation protocol. So, my current score for this paper is "marginally above the acceptance threshold", and I will adjust my score based on other reviews and authors' reply to my concerns.

- NFE and wall-clock time for AdjointDPM and the baselines is missing, so it is difficult to gauge the efficiency of AdjointDPM.
- A background section explaining and comparing other related backpropagation methods, such as DOODLE, FlowGrad, DEQ_DDIM in detail would help readers understand the position of AdjointDPM w.r.t. previous work. Specifically, this paper lacks a discussion of theoretical and practical advantages/disadvantages of AdjointDPM w.r.t previous work on backpropagation through diffusion sampling. For instance, the authors state DEQ-DDIM require the diffusion sampling process to have equilibrium points -- is this a significant drawback? Are there certain tasks where this equilibrium assumption do not hold, so AdjointDPM is applicable while DEQ-DDIM is not?

**Questions:**

See Weaknesses.

---

> ### Author Response · Authors · 2023-11-21
> **Reply to Reviewer CqXY - part I**
>
> Thank you for carefully reviewing this work and asking us very insightful questions! We have tried our best to answer your questions below.
>
>  > NFE and wall-clock time for AdjointDPM and the baselines are missing, so it is difficult to gauge the efficiency of AdjointDPM.
>
> **Reply:** For the analysis of NFE, please refer to the global response. The actual training time highly depends on the hardware configuration.
>
> When we finetune parameters of neural networks, for example, the stylization task, it takes around 18 hours to optimize in a A100 GPU. But once trained, this network can be used to generate any images with a certain style. The training time is acceptible in this case.
>
> For guided sampling tasks (such as vocabulary expansion, security auditing, and text embedding inversion), as the key time consumption is in traversing sampling steps and doing gradient backpropagation, here we present the wall-clock time required for a single optimization step. In these tasks, we run the code in a V100 GPU. For optimization of initial noises and text embedding inversion in Stable diffusion, it takes around 30 seconds per optimization step. For optimization of initial noises in ImageNet 256x256 generation, it takes around 5.3 seconds per optimization step.
>
> > A background section explaining and comparing other related backpropagation methods, such as DOODLE, FlowGrad, DEQ_DDIM in detail would help readers understand the position of AdjointDPM w.r.t. previous work. Specifically, this paper lacks a discussion of theoretical and practical advantages/disadvantages of AdjointDPM w.r.t previous work on backpropagation through diffusion sampling. For instance, the authors state DEQ-DDIM require the diffusion sampling process to have equilibrium points -- is this a significant drawback? Are there certain tasks where this equilibrium assumption do not hold, so AdjointDPM is applicable while DEQ-DDIM is not?
>  >
> **Reply:** We have presented part of comparsion with existing models, mainly including DOODL and FlowGrad in Appendix B. Here we have a brief comparison:
>
> 1. **Comparison with [1] End-to-End Diffusion Latent Optimization Improves Classifier Guidance**
>     DOODL in [1] optimizes the initial diffusion noise vectors w.r.t a model-based loss on images generated from the full-chain diffusion process. In their work, they obtain the gradients of loss w.r.t noise vectors by using invertible neural networks (INNs). There are two main differences between [1] and our work:
>   - While DOODL optimizes the initial diffusion noise vectors, our work optimizes related variables, including network parameters, initial noises and textual embeddings w.r.t a model-based loss on images generated from the full-chain diffusion process. Thus, we consider the broader cases of DOODL.
>   - In the calculation of gradients w.r.t initial noises, [1] uses the invertibility of EDICT, i.e., $x_0$ and $x_T$ are invertible. This method does not apply to the calculation of gradients w.r.t. the network parameters and textual embeddings as they share across the full-chain diffusion process. Finally, with regard to the memory consumption when calculating gradients with respect to the initial noise, our experimental results are as follows: We utilized the stable diffusion v1.4 checkpoint to run both the AdjointDPM and DOODL models on a V100 GPU (32GB memory). For the AdjointDPM method, backpropagating the gradients with respect to the initial noise required 19.63GB of memory. In comparison, the DOODL method consumed 23.5GB for the same operation.  Thus, our method is more efficient in terms of memory consumption. In terms of time consumption, DOODL relies on the invertibility of EDICT, resulting in identical computation steps for both the backward gradient calculation and the forward sampling process. However, our AdjointDPM methods have the flexibility to design adaptive steps, allowing for faster backward gradient calculation.

---

> > ### Author Response · Authors · 2023-11-21
> > **Reply to Reviewer CqXY - part II**
> >
> > 2. **Comparison with [2] FlowGrad: Controlling the Output of Generative ODEs With Gradients**
> >         FlowGrad efficiently backpropagates the output to any intermediate time steps on the ODE trajectory, by decomposing the backpropagation and computing vector Jacobian products. FlowGrad focuses on refining the ODE generation paths to the desired direction. This is different from our work, which focuses on the finetuning of related variables, including network parameters, textual embedding and initial noises of diffusion models for customization. Besides, FlowGrad methods also can not obtain the gradients of loss w.r.t. textual embeddings and neural variables as these variables share across the whole generation path. Then for the gradients w.r.t the latent variables, we could show the memory consumption of our methods is constant while they need to store the intermediate results. Since their method needs to store the intermediate results, in the paper, they present examples on Latent Diffusion Model (LDM) on FFHQ and LSUN Church, and Rectified Flow (RF) on CelebA-HQ, which are not large-scale diffusion models. Additionally, their officially released code for FlowGrad is only on pre-trained Rectified Flow [4], where the diffusion ODE path is a straight flow. Thus, when dealing with stable diffusion models, the use of their method may lead to inefficient memory utilization.
> >
> >
> >     [4] Liu, X., Gong, C., & Liu, Q. (2022). Flow straight and fast: Learning to generate and transfer data with rectified flow. arXiv preprint arXiv:2209.03003.
> >
> > 3. **Comparison with the paper [3] Deep Equilibrium Approaches to Diffusion Models (DEQ-DDIM)**
> >     Deep equilibrium approaches model the entire DDIM sampling chain as a joint, multivariate fixed point system. It solves all the equilibria simultaneously, which leads to faster convergence to final outputs. When doing model inversion (i.e., optimizing the initial noise), it directly calculates the gradients on the obtained fixed point. This method needs to store intermediate results, resulting in high memory consumption.
> >
> >     But for Adjoint method, it is general for all ODE models and suitable for different numerical solvers, including first-order (DDIM) and higher-order solvers. Besides, to obtain the gradients, adjoint sensitivity methods in our paper solve a backward ODE with constant memory. Thus, DEQ-DDIM is more time-efficient than AdjointDPM, but AdjointDPM is more memory-efficient than DEQ-DDIM.
> >
> >
> >     We conducted a comparison of the memory consumption between our method and DEQ when performing model inversion. Utilizing the official code from DEQ and the CIFAR10 checkpoint, we ran the model on a V100 GPU (32GB memory). When sampling an image using DEQ, the memory consumption reached 19GB, and the system exhausted its memory during the model inversion process.
> >
> >      For the AdjointDPM methods, we applied the stable diffusion v1.5 checkpoint for model inversion. The memory consumption for sampling an image was 10.3GB, and it required 19.63GB to perform the model inversion. As the stable diffusion v1.5 checkpoint is much larger than the CIFAR10 checkpoint, our method is shown to be much more memory efficient. Regarding computation time, we did not evaluate this aspect due to limitations in our GPU resources as it ran out of memory when we run the model inversion of DEQ.

---

> > > ### Comment · Reviewer_CqXY · 2023-11-21
> > >
> > > Thank you for the detailed feedback! After weighing the pros (practicality of AdjointDPM compared to [1,2,3], as discussed in the authors' feedback) and cons (lack of theoretical or methodological novelty, as this method is a straightforward application of the adjoint sensitivity method), I'm satisfied with my current score, which is already positive.

---

### Official Review · Reviewer_5NNi · 2023-11-01

**Soundness:** 3 good
**Presentation:** 3 good
**Contribution:** 2 fair
**Rating:** 6
**Confidence:** 4

**Summary:**

This work leverages the adjoint methods for optimizing the parameters and/or samples of diffusion ODEs under a given differentiable scalar-valued function. To efficiently solve the adjoint ODE, this work also leverages the expoenential integrators and introduce a change-of-variable formula to obtain a simpler ODE. Experiments show that the proposed method can be used for classifier-based sampling, adversarial sampling and stylization with a single reference.

**Strengths:**

- The proposed method is easy to understand and the writing is clean and easy to follow.
- The proposed adjoint method is novel to the diffusion model community and the combination with exponential integrator is useful.
- The studied topic is important to the field.

**Weaknesses:**

- Major:

  - The proposed method seems to be quite **inefficient** because it needs to optimize the model / sample at each specific task, while other guided sampling methods (e.g., classifier guidance or classifier-free guidance) do not. Note that the optimization procedure needs to solve the whole ODE at each training step, the training cost (i.e., total training time) seems to be quite expensive.

  - The proposed method cannot guarantee the property of diffusion models, i.e., the noise-pred network corresponds to the score functions, because it directly train the neural ODE. Thus, it may be hard to leverage the other properties of diffusion models, such as classifier / classifier-free guidance, and it may be hard to further use diffusion SDEs for better sample quality. Instead, other guided sampling methods introduce another guidance model at each time step, which is based on the score functions and thus can maintain the diffusion property.

- Minor:

  - Equation(9) is exactly the EDM sampler so it is not a new method. It should be compared and discussed in detail.

[1] Tero Karras, Miika Aittala, Timo Aila, and Samuli Laine. Elucidating the design space of diffusion-based generative models. In Advances in Neural Information Processing Systems, 2022.

**Questions:**

1. What is the training time for each experiment?

2. After training, can the model be used for classifier / classifier-free guidance and diffusion SDEs?

==========

Thanks for the detailed reply! After reading the authors' rebuttal, I raised my score to 6.

---

> ### Author Response · Authors · 2023-11-21
> **Reply to Reviewer 5NNi - part I**
>
> Thank you for carefully reviewing this work. We have tried our best to answer your questions below.
>
> > Weakness Major 1. The proposed method seems to be quite inefficient ... the training cost (i.e., total training time) seems to be quite expensive.
>
> **Reply:**  AdjointDPM can be used for both finetuning network weights and guided sampling.
>
> Regarding fine-tuning network weights, we only need to train the model **once** for a specific task. After training, we can use any type of sampling method to generate new results. The efficiency of finetuing is not an issue.
>
> On the other hand, in guided sampling (i.e., fine-tuning noisy samples during the sampling process and text embedding inversion), AdjointDPM is much more flexible than CG or CFG. We need **pairs** of data of image and guidance to train a classifier or CFG. In contrast, we do not need paired data in AdjointDPM since it supports any type of computable metrics defined on the generated results. Besides, it is not possible for CG or CFG to perform some tasks like security auditing or single image guided generation.
>
> > Weakness Major 2. The proposed method cannot guarantee the property of diffusion models..., which is based on the score functions and thus can maintain the diffusion property.
>
> > Question 2. After training, can the model be used for classifier / classifier-free guidance and diffusion SDEs?
>
> **Reply:** Thanks for raising this issue.
>
> **For finetuning of initial noises**, as long as we control the scale of perturbation on initial noises, the trained noises have little influence on the sampling process of diffusion models. Then for finetuning of text embedding, it also has little influence on score estimation as we just want to learn a proper text embedding for specific visual effects. Thus, CG/CFG and SDEs still apply in these two cases.
>
> In related experiments, besides the guidance from the loss function defined on final outputs, we still can use CG/CFG and SDEs in sampling. For example, in security auditing, we use CG to generate images with prescribed classes and find an initial noise which can mislead pretrained ImageNet classifier while visually similar to the original one. Besides, after finding this initial noise and text embedding, we could still use SDEs (such as DDPM sampler) to generate better-quality images. The results are added to Appendix.
>
> However, **when using AdjointDPM to finetune or train the parameters of diffusion networks**, the question of whether this operation will change the property of the underlying diffusion models is still largely open. However, we surmise that finetuning a diffusion model on a small amount of data for a few steps will still preserve the applicability of CG/CFG techniques. In practice, after finetuning the parameters of the Unet only for 10 epochs in stylization, we still use the CFG to guide the sampling with given text prompts under a guidance scale of 7.5. We also test the generation in SDE solvers and it works. The results are added to Appendix.
>
> Besides, we observe, in some other works that distill diffusion models based on Neural ODEs (e.g., consistency model, rectified flow), the resultant/distilled model lacks sampling diversity. This is a commonly-encountered problem concerning finetuning of diffusion models based on deterministic ODEs. In the future, we will explore techniques for effectively finetuing diffusion models and preserving their original properties.
>
> > Weakness Minor 1. Equation(9) is exactly the EDM sampler so it is not a new method. It should be compared and discussed in detail.
>
> **Reply:** The exponential integration technique has been used in many diffusion ODE-type solvers, including EDM [1], DPM-solver [2], DEIS [3]. Here our point is not to compare the difference on the exponential integration technique in different ODE solvers. We emphasize that AdjointDPM can retain comparable image generation quality with existing models (see FID results in Table 1), and at the same time, it can support obtaining the gradients by solving a backward ODE. Exploiting semi-linear accelerates both the forward and backward ODEs as there also exists a semi-linear structure in the backward ODE.
>
> [1] Karras, T., et al. (2022). Elucidating the design space of diffusion-based generative models.
>
> [2] Lu, C., et al. (2022). Dpm-solver: A fast ode solver for diffusion probabilistic model sampling in around 10 steps.
>
> [3] Zhang, Q., et al. (2022). Fast Sampling of Diffusion Models with Exponential Integrator.

---

> ### Author Response · Authors · 2023-11-21
> **Reply to Reviewer 5NNi - part II**
>
> > Question 1. What is the training time for each experiment?
>
> **Reply:** For the analysis of NFE, please refer to the global response. The actual training time highly depends on hardware configuration. When we finetune parameters of neural networks, for example, the stylization task, it takes around 18 hours to optimize in a A100 GPU. But once trained, this network can be used to generate any images with a certain style. The training time is acceptible in this case.
>
> For guided sampling tasks (such as vocabulary expansion, security auditing, and text embedding inversion), the key time consumption is in traversing sampling steps and doing gradient backpropagation, here we present the wall-clock time required for a single optimization step. In these tasks, we run the code in a V100 GPU. For optimization of initial noises and text embedding inversion in Stable Diffusion, it takes around 30 seconds per optimization step. For optimization of initial noises in ImageNet 256x256 generation, it takes around 5.3 seconds per optimization step.

---

> > ### Comment · Reviewer_5NNi · 2023-11-22
> > **Thank you for the detailed discussion!**
> >
> > I appreciate the authors' responses and I think it addressed my concerns, so I raised the score to 6.

---

### Official Review · Reviewer_6hcA · 2023-11-03

**Soundness:** 3 good
**Presentation:** 3 good
**Contribution:** 3 good
**Rating:** 6
**Confidence:** 5

**Summary:**

The paper proposes an interesting idea, named AdjointDPM, merging Diffusion Models and techniques from Neural ODE literature. The core offering of the paper is a way of backpropagating gradients of any loss computed using the output of a (trained) Diffusion model. Specifically the authors used the well-known Adjoint Backpropagation method from Neural ODE, which is a backprop algorithm with $\mathcal{O}(1)$ memory w.r.t the *discretization* of the ODE solver.

The authors applied their AdjointDPM method on three tasks that either require gradients w.r.t initial state $X_T$ of the reverse process, all intermediate states $\\{ X_t \\}\_{t=1}^T$ of the reverse process or the parameters $\theta$ of denoising model $\epsilon_{\theta}(\cdot)$. They showed good performance in terms of quantitative metrics and also showed qualitative results.

**Strengths:**

The proposal of the paper is overall good, theoretically sound and shown to have worked well.

- Theoretically, it makes sense to use the Adjoint method on the reverse ODE.
- The authors exploited the semi-linear nature of the ODE even in the Adjoint backprop, following DPM-Solver/DEIS.

**Weaknesses:**

- While the proposal is quite novel, one might still argue that it is not really necessary to use Adjoint Backprop. One can very well accomplish the same task by backprop-ing through the solver machinery (maybe by deceasing sampling steps and using better sampler), which of course, won’t be very efficient. So, at the end, it all boils down to compute/memory efficiency. While I understand the memory advantage, sadly, the paper barely talks anything about computational requirements of the method. BTW, Neural ODEs (and Adjoint Backprop) are known to be not very scalable.
- Experiments are okay-ish, but not really extensive. Qualitative samples are lacking in some experiments (e.g. vocabulary expansion). Also Vocabulary Expansion is shown for only two classes.
- No comparison or mention of methods that DO backprop through the ODE solver.

**Questions:**

I have the following questions for the authors.

- I am confused about section 3.4. That is not AdjointDPM — that is just unconditional generation with an already known sampler (DEIS, which used Adam-Bashforth) that exploits the semi-linear nature. What part of this is your contribution ? Am I missing something here ?
- In the “security auditing” application, what exactly is the guidance $L$ ? What is the meaning of “distance between a harmful prompt and a prediction score” ? Also, what exactly is the NSFW filter $f(\cdot)$ ? Can you provide more details please ?

Minor questions or suggestions:

- The unnumbered eq b/w Eq. 5 & 6 — what is the meaning of that line (”Similarly, for $\theta$, we can regard ..”) ?
- “adjoint state $\mathbf{a}(t) = ..$, which represents how the loss depends on the state ..” → “adjoint state $\mathbf{a}(t)$ = .., which represents how the loss **changes w.r.t the** state ..”.
- The AdjointDPM eq. 8 shows gradient w.r.t $t$ — is it acutally used anywhere ?
- Why is it called “vocabulary expansion” ? I still don’t get it.
- “Security Auditing” is just fancy name for Adversarial Samples ?

---

> ### Author Response · Authors · 2023-11-21
> **Reply to Reviewer 6hcA**
>
> Thank you for carefully reviewing this work and asking us very insightful questions! We have tried our best to answer your questions below.
> > Weakness 1. While the proposal is quite novel, one might still argue that it is not really necessary to use Adjoint Backprop....While I understand the memory advantage, sadly, the paper barely talks anything about the computational requirements of the method. BTW, Neural ODEs (and Adjoint Backprop) are known to be not very scalable.
> > Weakness 3. No comparison or mention of methods that DO backprop through the ODE solver.
> >
> **Reply:**
> - When we try to finetune the state-of-the-art text-to-image diffusion model (e.g., Stable Diffusion), it is nearly infeasible to do direct backpropagation. Stable Diffusion usually requires more than 25 steps to produce a high-quality image, however even V100-GPU cannot back-propagate gradients throughout 5 times of iterations (31913MiB / 32768MiB), let alone 25 iterations.
> - For the memory consumption, NFE analysis, and scalability, please refer to Global Response.
>
> > Weakness 2. Experiments are okay-ish, but not really extensive. Qualitative samples are lacking in some experiments (e.g. vocabulary expansion). Also Vocabulary Expansion is shown for only two classes.
>
>   **Reply:** Thanks for raising this point. We extend our results comparison to 90 classes in Standford dogs Dataset. We compute the change of FID relative to the original Stable Diffusion generations for comparison. The results are shown in the following table. As there is limited time to run DOODL's result for us, we use the value reported in [1]. We also did not extend the comparison to more classes on the Birds dataset. Complete quantitative results will be added to the final version of the manuscript in Table 2. More qualitative results are added in Appendix.
>
>   | Dataset   | DOODL | AdjointDPM|
> | -------- |  ------- |  ------- |
> | Dogs  | -5.3% | -7.1% |
>
> [1] Bram Wallace, et al. End-to-end diffusion latent optimization improves classifier guidance.
>
> > Q1. I am confused about section 3.4. That is not AdjointDPM — that is just unconditional generation with an already known sampler (DEIS, which used Adam-Bashforth) that exploits the semi-linear nature. What part of this is your contribution? Am I missing something here?
>
> **Reply:**  Sorry for the confusion. The exponential integration technique has been used in many diffusion ODE-type solvers, including EDM [1], DPM-solver [2], DEIS [3]. Here our point is not to compare the difference on the exponential integration technique in different ODE solvers. We emphasize that AdjointDPM can retain comparable image generation quality with existing models (see FID results in Table 1), and at the same time, it can support obtaining the gradients by solving a backward ODE. Exploiting semi-linear accelerates both the forward and backward ODEs as there also exists a semi-linear structure in the backward ODE.
>
> [1] Karras, T., et al. (2022). Elucidating the design space of diffusion-based generative models.
>
> [2] Lu, C., et al. (2022). Dpm-solver: A fast ode solver for diffusion probabilistic model sampling in around 10 steps.
>
> [3] Zhang, Q., et al. (2022). Fast Sampling of Diffusion Models with Exponential Integrator.
>
> > Q2. In the “security auditing” application, what exactly is the guidance? What is the meaning of “distance between a harmful prompt and a prediction score”? Also, what exactly is the NSFW filter $f()$? Can you provide more details please?
>
> **Reply:** Security auditing aims to search for adversarial initial noises $\mathbf{x}_T$ that will evolve into harmful images and bypass the filter. In the paper, we present two examples: the first one sets the filter to be a pretrained ImageNet classifier (in the experiment, we use the pretrained Resnet50) and the second one is the Not Safe For Work (NSFW) Filter [2] in Stable Diffusion.
>
> For the ImageNet classifier, the guidance is set to be the cross entropy loss between the classification results of the generated images and any class that is not equal to the classification results (we choose this class randomly). For the NSFW filter, we set the guidance to be a weighted combination of cosine similarities between CLIP image features of the generated unsafe image and CLIP text features of some harmful concepts [2]. Usually in an NSFW filter, when the similarity of the generated image to harmful features is greater than a threshold, the generated images will be filtered. Thus, we minimize the similarity to help the generated unsafe images skip the filter.
>
> [2] Schramowski, et al. (2023). Safe latent diffusion: Mitigating inappropriate degeneration in diffusion models.

---

> > ### Author Response · Authors · 2023-11-21
> > **Reply to Reviewer 6hcA - Reply to Minor questions**
> >
> > >  Q3. The unnumbered eq b/w Eq. 5 & 6 — what is the meaning of that line (”Similarly, for, we can regard ..”)?
> >
> > **Reply:** Eq. (5) introduces the adjoint state corresponding to $\mathbf{x}$. The unnumbered eq b/w Eqs. (5) and (6) introduce the augmented states $\theta$ and $t$ and their corresponding adjoint states.
> >
> > > Q4. “adjoint state, which represents how the loss depends on the state ..” → “adjoint state = .., which represents how the loss changes w.r.t the state ..”.
> >
> > **Reply:** Thanks for pointing this out. We will correct it in the paper.
> >
> > > Q5. The AdjointDPM eq. 8 shows gradient w.r.t $t$— is it actually used anywhere?
> >
> > **Reply:** No. Gradient w.r.t $t$ is not used. Here we just introduce it for completeness in the introduction of the adjont method.
> >
> > > Q6. Why is it called “vocabulary expansion”? I still don’t get it.
> >
> > **Reply:** “Vocabulary expansion” is a task where we use fine-grained visual classification (FGVC) models to guide the diffusion generation process. This can help refine the vocabulary of a pretrained diffusion models such that it can generate a specific class learned by an FGVC classifier. In detail, the majority of concepts present in FGVC models are rare or non-existent in the training data of diffusion models and as such, the pretrained diffusion models cannot generate accurate images for them. Then by using AdjointDPM to guide the sampling procedure, the text-to-image model can generate the target object. This is equivalent to expanding the vocabulary (knowledge base) of the text-to-image model.
> >
> > >Q7. “Security Auditing” is just a fancy name for Adversarial Samples?
> >
> > **Reply:** Yes. But in this task, we want to emphasize that there exists a subset of random noise that will fail the content-moderation system of diffusion models. It triggers an alert that we have to design more robust and secure AI generative systems.

---

> > > ### Comment · Reviewer_6hcA · 2023-11-22
> > > **Thanks for the rebuttal**
> > >
> > > Thanks to the authors for the rebuttal. They clarified a most of my doubts. They provided more experiments, and most importantly some computational analysis.
> > >
> > > While I still think the computations are relatively costly, I would like to re-terate my stand that the paper is conceptually/theoretically good. Weighting pros and cons, I think I would like to stick to my score, which I think is a right balance.

---

### Author Response · Authors · 2023-11-21
**Global Response**

Reviewers concern about the efficiency of AdjointDPM. We discuss the time consumption (NFE), GPU memory, and the scalability as follows:
- **Number of Neural Function Evaluation (NFE) Analysis**:
    AdjointDPM can be used for finetuing network's weights and optimizing the noisy states. For both cases, assuming $T$ sampling steps, NFEs for each optimization update is $2T$ ($T$ for forward generation, $T$ for backward gradient calculation). Regarding the guided sampling (i.e., optimizing the noisy states), other existing approaches like DOODL[1] and FlowGrad[2], also cannot avoid traversing all sampling steps. Their NFEs of these methods are also $2T$.

- **Memory consumption for AdjointDPM:**
  Memory consumption depends on the scale of optimizable parameters.

  - For stylization, we finetune of parameters of Unet under the Stable Diffusion v1.4 models for 512x512 image generation. It takes around 65GB of memory.
  - For guided sampling tasks, we finetune the initial noises and textual embedding, respectively. The memory consumption shown as follows:
     - **Finetuning initial noise on Stable Diffusion v1.5**: The memory consumption of AdjointDPM is 19.63GB. For comparsion, DOODL[1] consumes 23.5GB. If we do direct backpropagation, using 5 sampling steps is nearly out of memory (31913MiB / 32768MiB).
     - **Finetuning textual embedding on Stable Diffusion v1.5**: The memory consumption of AdjointDPM is 14.64GB. If we do direct backpropagation, using 6 sampling steps is nearly out of memory (30839MiB / 32768MiB).
- **The scalability of AdjointDPM**
    The scalability of adjoint method may be a real issue when training a large neural ODE model on large-scale datasets. However, in our case, we mainly focus on using the AdjointDPM for **"finetuning"** the parameters of diffusion models. This means we only update the parameters on very few numbers of data. Thus, the scalability of AdjointDPM may not be a practical issue. Specifically,
    1. For fine-tuning parameter tasks, we only optimize for small epochs (~10 in our experiment). There exist techniques for further improvement of efficiency, such as truncating backpropagation steps, and reducing the scale of parameters using LoRA.
    2. For guided sampling, we provide a general method to obtain the gradients of states. In some real applications, such as object-guided generation, it's not necessary to traverse all the sampling steps. We could only optimize the several middle noisy samples for synthesizing images of certain objects (for example, vocabulary expansion in our experiments). As such, the computational cost will significantly decrease.

---

### Author Response · Authors · 2023-11-22

Dear reviewers,

Thank you for your effort in reviewing our paper and offering valuable comments. We have provided corresponding responses, which we believe have covered your concerns. As the discussion period is close to its end, we would appreciate your feedback on whether or not our responses adequately address your concerns. Please let us know if you still have any concerns that we can resolve.

---

### Meta-Review · Area_Chair_JfWA · 2023-12-05

**Metareview:**

This paper proposes AdjointDPM, an adjoint sensitivity method for gradient backpropagation in diffusion methods. The formulation is novel for me and most of the reviewers.

The proposed method is memory-efficient compared to naive BP and applicable to the optimization of parameters, conditions, and noisy states. Empirical results on vocabulary expansion, security auditing, and style transfer show that the proposed methods outperform baselines (mainly on benchmarks conducted by the authors).

All reviewers reached an agreement (with the same rating of 6). My major concerns include: 1. the improvement in memory efficiency is not that significant to more relevant papers, e.g. 19.63GB for the proposed method and 23.24GB for DOODL. 2. the methods are mainly evaluated on benchmarks conducted by the authors. Such issues may limit the significance of the paper.

Currently, I tend to accept it as a poster but I won't mind if it gets rejected.

**Justification For Why Not Higher Score:**

My major concerns include: 1. the improvement in memory efficiency is not that significant to more relevant papers, e.g. 19.63GB for the proposed method and 23.24GB for DOODL. 2. the methods are mainly evaluated on benchmarks conducted by the authors. Such issues may limit the significance of the paper.

**Justification For Why Not Lower Score:**

All reviewers reached an agreement (with the same rating of 6).

---

### Decision · Program_Chairs · 2024-01-16

Accept (poster)